# SIMILARITY SEARCH FOR EFFICIENT ACTIVE LEARNING AND SEARCH OF RARE CONCEPTS

## ABSTRACT

Many active learning and search approaches are intractable for industrial settings with billions of unlabeled examples. Existing approaches, such as uncertainty sampling or information density, search globally for the optimal examples to label, scaling linearly or even quadratically with the unlabeled data. However, in practice, data is often heavily skewed; only a small fraction of collected data will be relevant for a given learning task. For example, when identifying rare classes, detecting malicious content, or debugging model performance, positive examples can appear in less than 1% of the data. In this work, we exploit this skew in large training datasets to reduce the number of unlabeled examples considered in each selection round by only looking at the nearest neighbors to the labeled examples. Empirically, we observe that learned representations can effectively cluster unseen concepts, making active learning very effective and substantially reducing the number of viable unlabeled examples. We evaluate several selection strategies in this setting on three large-scale computer vision datasets: ImageNet, OpenImages, and a proprietary dataset of 10 billion images from a large internet company. For rare classes, active learning methods need as little as 0.31% of the labeled data to match the average precision of full supervision. By limiting the selection strategies to the immediate neighbors of the labeled data as candidates for labeling, we process as little as 0.1% of the unlabeled data while achieving similar reductions in labeling costs as the traditional global approach. This process of expanding the candidate pool with the nearest neighbors of the labeled set can be done efficiently and reduces the computational complexity of selection by orders of magnitude.

## 1 INTRODUCTION

Large-scale unlabeled datasets contain millions or billions of examples spread over a wide variety of underlying concepts (Chelba et al., 2013; Zhu et al., 2015; Zhang et al., 2015; Wan et al., 2019; Russakovsky et al., 2015; Kuznetsova et al., 2020; Thomee et al., 2016; Abu-El-Haija et al., 2016; Caesar et al., 2019; Lee et al., 2019). Often, these massive datasets skew towards a relatively small number of common concepts, such as cats, dogs, and people (Liu et al., 2019; Zhang et al., 2017; Wang et al., 2017; Van Horn & Perona, 2017). Rare concepts, such as harbor seals, may only appear in a small fraction of the data (less than 1%). However, in many settings, performance on these rare concepts is critical. For example, harmful or malicious content may comprise a small percentage of user-generated content, but it can have an outsize impact on the overall user experience (Wan et al., 2019). Similarly, when debugging model behavior for safety-critical applications like autonomous vehicles, or when dealing with representational biases in models, obtaining data that captures rare concepts allows machine learning practitioners to combat blind spots in model performance (Karpathy, 2018; Holstein et al., 2019; Ashmawy et al., 2019; Karpathy, 2020). Even a simple prediction task like stop sign detection can be challenging given the diversity of real-world data. Stop signs may appear in a variety of conditions (e.g., on a wall or held by a person), be heavily occluded, or have modifiers (e.g., "Except Right Turns") (Karpathy, 2020). While large-scale datasets are core to addressing these issues, finding the relevant examples for these long-tail tasks is challenging.

Active learning and search have the potential to automate the process of identifying these rare, high value data points significantly, but existing methods become intractable at this scale. Specifically, the goal of active learning is to reduce the cost of labeling (Settles, 2012). To this end, the learning algorithm is allowed to choose which data to label based on uncertainty (e.g., the entropy of

predicted class probabilities) or other heuristics (Settles, 2011; 2012; Lewis & Gale, 1994). Active search is a sub-area focused on finding positive examples in skewed distributions (Garnett et al., 2012). Because of a concentrated focus on labeling costs, existing techniques, such as uncertainty sampling (Lewis & Gale, 1994) or information density (Settles & Craven, 2008), perform multiple selection rounds and iterate over the entire unlabeled data to identify the optimal example or batch of examples to label and scale linearly or even quadratically with the size of the unlabeled data. Computational efficiency is becoming an impediment as the size of datasets and model complexities have increased (Amodei & Hernandez, 2018). Recent work has tried to address this problem with sophisticated methods to select larger and more diverse batches of examples in each selection round and reduce the total number of rounds needed to reach the target labeling budget (Sener & Savarese, 2018; Kirsch et al., 2019; Coleman et al., 2020; Pinsler et al., 2019; Jiang et al., 2018). Nevertheless, these approaches still scan over all of the examples to find the optimal examples to label in each round and can be intractable for large-scale unlabeled datasets. For example, running a single inference pass over 10 billion images with ResNet-50 (He et al., 2016) would take 38 exaFLOPs.

In this work, we propose Similarity search for Efficient Active Learning and Search (SEALS) to restrict the candidates considered in each selection round and vastly reduce the computational complexity of active learning and search methods. Empirically, we find that learned representations from pre-trained models can effectively cluster many unseen and rare concepts. We exploit this latent structure to improve the computational efficiency of active learning and search methods by only considering the nearest neighbors of the currently labeled examples in each selection round. This can be done transparently for many selection strategies making SEALS widely applicable. Finding the nearest neighbors for each labeled example in unlabeled data can be performed efficiently with sublinear retrieval times (Charikar, 2002) and sub-second latency on billion-scale datasets (Johnson et al., 2017) for approximate approaches. While constructing the index for similarity search requires at least a linear pass over the unlabeled data, this computational cost is effectively amortized over many selection rounds or other applications. As a result, our SEALS approach enables selection to scale with the size of the labeled data rather than the size of the unlabeled data, making active learning and search tractable on datasets with billions of unlabeled examples.

We empirically evaluated SEALS for both active learning and search on three large scale computer vision datasets: ImageNet (Russakovsky et al., 2015), OpenImages (Kuznetsova et al., 2020), and a proprietary dataset of 10 billion images from a large internet company. We selected 611 concepts spread across these datasets that range in prevalence from 0.203% to 0.002% (1 in 50,000) of the training examples. We evaluated three selection strategies for each concept: max entropy uncertainty sampling (Lewis & Gale, 1994), information density (Settles & Craven, 2008), and most-likely positive (Warmuth et al., 2002; 2003; Jiang et al., 2018). Across datasets, selection strategies, and concepts, SEALS achieved similar model quality and nearly the same recall of the positive examples as the baseline approaches, while improving the computational complexity by orders of magnitude. On ImageNet with a budget of 2,000 binary labels per concept (~0.31% of the unlabeled data), all baseline and SEALS approaches were within 0.011 mAP of full supervision and recalled over 50% of the positive examples. On OpenImages, SEALS reduced the candidate pool to 1% of the unlabeled data on average while remaining within 0.013 mAP and 0.1% recall of the baseline approaches. On the proprietary dataset with 10 billion images, SEALS needed an even smaller fraction of the data, about 0.1%, to match the baseline, which allowed SEALS to run on a single machine rather than a cluster. To the best of our knowledge, no other works have performed active learning at this scale. We also applied SEALS to the NLP spoiler detection dataset Goodreads (Wan et al., 2019), where it achieved the same recall as the baseline approaches while only considering less than 1% of the unlabeled data. Together, these results demonstrate that SEALS' improvements to computational efficiency make active learning and search tractable for even billion-scale datasets.

## 2 RELATED WORK

**Active learning**'s iterative retraining combined with the high computational complexity of deep learning models has led to significant work on computational efficiency (Sener & Savarese, 2018; Kirsch et al., 2019; Pinsler et al., 2019; Coleman et al., 2020; Yoo & Kweon, 2019; Mayer & Timofte, 2020; Zhu & Bento, 2017). One branch of recent work has focused on selecting large batches of data to minimize the amount of retraining and reduce the number of selection rounds necessary to reach a target budget (Sener & Savarese, 2018; Kirsch et al., 2019; Pinsler et al., 2019). These

approaches introduce novel techniques to avoid selecting highly similar or redundant examples and ensure the batches are both informative and diverse. In comparison, our work aims to reduce the number of examples considered in each selection round and complements existing work on batch active learning. Many of these approaches sacrifice computational complexity to ensure diversity, and their selection methods can scale quadratically with the size of the unlabeled data. Combined with our method, these selection methods scale with the size of the labeled data rather than the unlabeled data. Outside of batch active learning, other work has tried to improve computational efficiency by either using much smaller models as cheap proxies during selection (Yoo & Kweon, 2019; Coleman et al., 2020) or by generating examples (Mayer & Timofte, 2020; Zhu & Bento, 2017). Using a smaller model reduces the amount of computation per example, but unlike our approach, it still requires making multiple passes over the entire unlabeled pool of examples. The generative approaches (Mayer & Timofte, 2020; Zhu & Bento, 2017), however, enable sub-linear runtime complexity like our approach. Unfortunately, they struggle to match the label-efficiency of traditional approaches because the quality of the generated examples is highly variable.

**Active search** is a sub-area of active learning that focuses on highly skewed class distributions (Garnett et al., 2012; Jiang et al., 2017; 2018; 2019). Rather than optimizing for model quality, active search aims to find as many examples from the minority class as possible. Prior work has focused on applications such as drug discovery, where the dataset sizes are limited, and labeling costs are exceptionally high. Our work similarly focuses on skewed distributions. However, we consider novel active search settings in image and text where the available unlabeled datasets are much larger, and computational efficiency is a significant bottleneck.

$k$ **nearest neighbor ($k$-NN)** classifiers are popular models in active learning and search because they do not require an explicit training phase (Joshi et al., 2012; Wei et al., 2015; Garnett et al., 2012; Jiang et al., 2017; 2018). The prediction and score for each unlabeled example can be updated immediately after each new batch of labels. In comparison, our SEALS approach uses $k$-NN algorithms for similarity search to create and expand the candidate pool and not as a classifier. This is an important but subtle difference. While prior work avoids expensive training by using $k$-NN classifiers, these approaches still require evaluating all of the unlabeled examples, which can still be prohibitively expensive on large-scale datasets like the ones we consider here. SEALS targets the selection phase rather than training, presenting a novel and complementary approach.

## 3 METHODS

In this section, we outline the problems of active learning (Section 3.1) and search (Section 3.2) formally as well as the selection methods we accelerate using SEALS. For both, we examine the pool-based setting, where all of the unlabeled data is available at once, and examples are selected in batches to improve computational efficiency, as mentioned above. Then in Section 3.3, we describe our SEALS approach and how it further improves computational efficiency in both settings.

### 3.1 ACTIVE LEARNING

Pool-based active learning is an iterative process that begins with a large pool of unlabeled data $U = \{\mathbf{x}_1, \dots, \mathbf{x}_n\}$. Each example is sampled from the space $\mathcal{X}$ with an unknown label from the label space $\mathcal{Y} = \{1, \dots, C\}$ as $(\mathbf{x}_i, y_i)$. We additionally assume a feature extraction function $G_z$ to embed each $\mathbf{x}_i$ as a latent variable $G_z(\mathbf{x}_i) = \mathbf{z}_i$ and that the $C$ concepts are unequally distributed. Specifically, there are one or more valuable rare concepts $R \subset C$ that appear in less than 1% of the unlabeled data. For simplicity, we frame this as $|R|$ binary classification problems solved independently rather than 1 multi-class classification problem with $|R|$ concepts. Initially, each rare concept has a small number of positive examples and several negative examples that serve as a labeled seed set $L_r^0$. The goal of active learning is to take this seed set and select up to a budget of $T$ examples to label that produce a model $A_r^T$ that achieves low error. For each round $t$ in pool-based active learning, the most informative examples are selected according to the selection strategy $\phi$ from a pool of candidate examples $\mathcal{P}_r$ in batches of size $b$ and labeled, as shown in Algorithm 1.

For the baseline approach, $\mathcal{P}_r = \{G_z(\mathbf{x}) \mid \mathbf{x} \in U\}$, meaning that all the unlabeled examples are considered to find the global optimal according to $\phi$. Between each round, the model $A_r^t$ is trained on all of the labeled data $L_r^t$, allowing the selection process to adapt.

In this paper, we considered **max entropy (MaxEnt)** uncertainty sampling (Lewis & Gale, 1994):

$$\phi_{\text{MaxEnt}}(\mathbf{z}) = -\sum_{\hat{y}} P(\hat{y}|\mathbf{z}; A_r) \log P(\hat{y}|\mathbf{z}; A_r)$$

and **information density (ID)** (Settles & Craven, 2008):

$$\phi_{\text{ID}}(\mathbf{z}) = \phi_{\text{MaxEnt}}(\mathbf{z}) \times \left( \frac{1}{|\mathcal{P}_r|} \sum_{\mathbf{z}_p \in \mathcal{P}_r} \text{sim}(\mathbf{z}, \mathbf{z}_p) \right)^{\beta}$$

where $\text{sim}(\mathbf{z}, \mathbf{z}_p)$ is the cosine similarity of the embedded examples and $\beta = 1$. Note that for binary classification, max entropy is equivalent to least confidence and margin sampling, which are also popular criteria for uncertainty sampling (Settles, 2009). While max entropy uncertainty sampling only requires a linear pass over the unlabeled data, ID scales quadratically with $|U|$ because it weights each example's informativeness by its similarity to all other examples. To improve computational performance, the average similarity score for each example can be cached after the first selection round, so subsequent rounds scale linearly. This optimization only works when $G_z$ is fixed and would not apply to dynamic similarity calculations like those in Sener & Savarese (2018).

We explored the greedy k-centers approach from Sener & Savarese (2018) but found that it never outperformed random sampling for our experimental setup. Unlike MaxEnt and ID, k-centers does not consider the predicted labels. It tries to achieve high coverage over the entire candidate pool, of which rare concepts make up a small fraction by definition, making it ineffective for our setting.

| **Algorithm 1** BASELINE APPROACH | **Algorithm 2** SEALS APPROACH |
|---|---|
| **Input:** unlabeled data $U$, labeled seed set $L_r^0$, feature extractor $G_z$, selection strategy $\phi(\cdot)$, batch size $b$, labeling budget $T$ | **Input:** unlabeled data $U$, labeled seed set $L_r^0$, feature extractor $G_z$, selection strategy $\phi(\cdot)$, batch size $b$, labeling budget $T$, $k$-nearest neighbors implementation $\mathcal{N}(\cdot, \cdot)$ |
| 1: $\mathcal{L}_r = \{(G_z(\mathbf{x}), y) \mid (\mathbf{x}, y) \in L_r^0\}$ | 1: $\mathcal{L}_r = \{(G_z(\mathbf{x}), y) \mid (\mathbf{x}, y) \in L_r^0\}$ |
| 2: $\mathcal{P}_r = \{G_z(\mathbf{x}) \mid \mathbf{x} \in U \text{ and } (\mathbf{x}, \cdot) \notin L_r^0\}$ | 2: $\mathcal{P}_r = \cup_{(\mathbf{z}, y) \in \mathcal{L}_r} \mathcal{N}(\mathbf{z}, k)$ |
| 3: **repeat** | 3: **repeat** |
| 4: $\quad A_r = \text{train}(\mathcal{L}_r)$ | 4: $\quad A_r = \text{train}(\mathcal{L}_r)$ |
| 5: $\quad$ **for** 1 to $b$ **do** | 5: $\quad$ **for** 1 to $b$ **do** |
| 6: $\quad\quad \mathbf{z}^* = \arg\max_{\mathbf{z} \in \mathcal{P}_r} \phi(\mathbf{z})$ | 6: $\quad\quad \mathbf{z}^* = \arg\max_{\mathbf{z} \in \mathcal{P}_r} \phi(\mathbf{z})$ |
| 7: $\quad\quad \mathcal{L}_r = \mathcal{L}_r \cup \{(\mathbf{z}^*, \text{label}(\mathbf{x}^*))\}$ | 7: $\quad\quad \mathcal{L}_r = \mathcal{L}_r \cup \{(\mathbf{z}^*, \text{label}(\mathbf{x}^*))\}$ |
| 8: $\quad\quad \mathcal{P}_r = \mathcal{P}_r \setminus \{\mathbf{z}^*\}$ | 8: $\quad\quad \mathcal{P}_r = (\mathcal{P}_r \setminus \{\mathbf{z}^*\}) \cup \mathcal{N}(\mathbf{z}^*, k)$ |
| 9: $\quad$ **end for** | 9: $\quad$ **end for** |
| 10: **until** $|\mathcal{L}_r| = T$ | 10: **until** $|\mathcal{L}_r| = T$ |

## 3.2 ACTIVE SEARCH

Active search is closely related to active learning, so much of the formalism from Section 3.1 carries over. The critical difference is that rather than selecting examples to label that minimize error, the goal of active search is to maximize the number of examples from the target concept $r$, expressed with the natural utility function $u(L_r) = \sum_{(\mathbf{x}, y) \in L_r} \mathbb{1}\{y = r\}$. As a result, different selection strategies are favored, but the overall algorithm is the same as Algorithm 1.

In this paper, we consider an additional selection strategy to target the active search setting, **most-likely positive (MLP)** (Warmuth et al., 2002; 2003; Jiang et al., 2018):

$$\phi_{\text{MLP}}(\mathbf{z}) = P(r|\mathbf{z}; A_r)$$

Because active learning and search are similar, we evaluate all the selection criteria from Sections 3.1 and 3.2 in terms of both the error the model achieves and the number of positive examples.

### 3.3 SIMILARITY SEARCH FOR EFFICIENT ACTIVE LEARNING AND SEARCH (SEALS)

In this work, we propose SEALS to accelerate the inner loop of active learning and search by restricting the candidate pool of unlabeled examples. To apply SEALS, we use an efficient method for similarity search of the embedded examples (Charikar, 2002; Johnson et al., 2017) and make two modifications to the baseline approach, as shown in Algorithm 2:

1. The candidate pool $\mathcal{P}_r$ is restricted to the nearest neighbors of the labeled examples.
2. After every example is selected, we find its $k$ nearest neighbors and update $\mathcal{P}_r$.

**Computational savings.** Restricting the candidate pool $\mathcal{P}_r$ to the $k$ nearest neighbors of the labeled examples means we only apply the selection strategy to at most $k|L_r|$ examples. This can be done transparently for many selection strategies making it applicable to a wide range of active learning and search methods, even beyond the ones considered here. Finding the $k$ nearest neighbors for each newly labeled example adds overhead, but this can be calculated efficiently with sublinear retrieval times (Charikar, 2002) and sub-second latency on billion-scale datasets (Johnson et al., 2017) for approximate approaches. As a result, the computational complexity of each selection round scales with the size of the labeled dataset rather than the unlabeled dataset. Excluding the retrieval times for the $k$ nearest neighbors, the computational savings from SEALS are directly proportional to the pool size reduction for $\phi_{\text{MaxEnt}}$ and $\phi_{\text{MLP}}$, which is lower bounded by $|U|/k|L_r|$. For $\phi_{\text{ID}}$, the average similarity score for each example only needs to be computed once when the example is first selected. This caching means the first round scales quadratically with $|U|$ and subsequent rounds scale linearly for the baseline approach. With SEALS, each selection round scales according to $O((1 + bk)|\mathcal{P}_r|)$ because the similarity scores are calculated as examples are selected rather than all at once. The resulting computational savings of SEALS varies with the labeling budget $T$ as the upfront cost of the baseline amortizes. Nevertheless, for large-scale datasets with millions or billions of examples, performing that first quadratic round for the baseline is prohibitively expensive.

**Index construction.** Generating the embeddings and indexing the data can be expensive and slow, requiring at least a linear pass over the unlabeled data. However, this cost is effectively amortized over many selection rounds, concepts, or other applications. Similarity search is a critical workload for information retrieval and powers many applications, including recommendation. Increasingly, embeddings from deep learning models are being used (Babenko et al., 2014; Babenko & Lempitsky, 2016; Johnson et al., 2017). As a result, the embeddings and index can be generated once using a generic model trained in a weak-supervision or self-supervision fashion and reused, making our approach just one of many applications using the index. Alternatively, if the data has already been passed through a predictive system (for example, to tag or classify uploaded images), the embedding could be captured and indexed at inference to avoid additional costs.

## 4 RESULTS

We applied SEALS to three selection strategies and performed active learning and search on three datasets: ImageNet (Russakovsky et al., 2015), OpenImages (Kuznetsova et al., 2020), and a proprietary dataset of 10 billion images. Section 4.1 details the experimental setup for each dataset and the inputs used for both the baseline approach (Algorithm 1) and our proposed method, SEALS (Algorithm 2). Sections 4.2 and 4.3 present the empirical results for active learning and search. Section 4.4 explores the structure of the concepts through the nearest neighbor graphs and embeddings.

Across selection strategies, datasets, and concepts, SEALS using ResNet-50 (He et al., 2016) embeddings performed similarly to the baseline while only considering a fraction of the unlabeled data $U$ in the candidate pool for each concept $\mathcal{P}_r$. For MLP and MaxEnt, the smaller candidate pool from SEALS sped-up the selection runtime by over $180\times$ on OpenImages. This allowed us to run active learning and search efficiently on an industrial scale dataset with 10 billion images. The improvements were even larger for information density. On ImageNet, SEALS dropped the time for the first selection round from over 75 minutes to 1.5 seconds, over a $3000\times$ improvement. On OpenImages, the baseline for information density ran for over 24 hours without completing a single round, while SEALS took less than 3 minutes to perform 19 rounds. We observed similar results with self-supervised embeddings using SimCLR (Chen et al., 2020) for ImageNet (Appendix A.3) and using SentenceBERT (Reimers & Gurevych, 2019) for Goodreads spoiler detection (Appendix A.8).

Table 1: Summary of datasets

| | Number of Concepts ($|R|$) | Embedding Model ($G_z$) | Number of Examples ($|U|$) | Fraction Positive $\left(\frac{\sum 1\{y=r\}}{|U|}\right)$ |
|---|---|---|---|---|
| ImageNet (Russakovsky et al., 2015) | 450 | ResNet-50 (He et al., 2016) (500 classes) | 639,906 | 0.114-0.203% |
| OpenImages (Kuznetsova et al., 2020) | 153 | ResNet-50 (He et al., 2016) (1000 classes) | 6,816,296 | 0.002-0.088% |
| 10B images (proprietary) | 8 | ResNet-50 (He et al., 2016) (1000 classes) | 10,094,719,767 | - |

## 4.1 EXPERIMENTAL SETUP

Across all datasets and selection strategies, we followed the same general procedure for both active learning and search. Because we are interested in rare concepts, we kept the number of initial positive examples small. We evaluated three settings, with 5, 20, and 50 positives, but only included the results with the smallest size in this section. The others are shown in Appendix A.1. For each setting, negative examples were randomly selected at a ratio of 19 negative examples to every positive example to form the seed set $L_r^0$. The slightly higher number of negatives in the initial seed set improved average precision on the validation set across all three datasets. The batch size $b$ for each selection round was the same as the size of the initial seed set. For the seed set of 5 positive and 95 negative examples shown below, $b$ was 100, and the labeling budget $T$ was 2,000 examples.

As the binary classifier for each concept $A_r$, we used logistic regression trained on the embedded examples. For active learning, we calculated average precision on the test data for each binary concept classifier after each selection round. For active search, we count the number of positive examples labeled so far. We take the mean average precision (mAP) and number of positives across concepts, run each experiment 5 times, and report the mean and standard deviation.

For similarity search, we used locality-sensitive hashing (LSH) (Charikar, 2002) implemented in Faiss (Johnson et al., 2017) with Euclidean distance for all datasets aside from the 10 billion images dataset. This simplified our implementation, so the index could be created quickly and independently for each concept and configuration, allowing experiments to run in parallel trivially. However, retrieval times for this approach were not as fast as Johnson et al. (2017) and made up a larger part of the overall active learning loop. In practice, the search index can be heavily optimized and tuned for the specific data distribution, leading to computational savings closer to the improvements described in Section 3.3 and differences in the "Selection" portion of the runtimes in Table 2.

We split the data, selected concepts, and created embeddings as detailed below and summarized in Table 1. Note that our approach does not constrain the choice of $G_z$, which allows for many network architectures. As representations continue to improve with new self-supervision, generative, or transfer learning techniques, SEALS is still applicable and performance will also likely improve.

**ImageNet** (Russakovsky et al., 2015) has 1.28 million training images spread over 1000 classes. To simulate rare concepts, we split the data in half, using 500 classes to train the feature extractor $G_z$ and treating the other 500 classes as unseen concepts. For $G_z$, we used ResNet-50 but added a bottleneck layer before the final output to reduce the dimension of the embeddings to 256. We kept all of the other training hyperparameters the same as in He et al. (2016). We extracted features from the bottleneck layer and applied $l^2$ normalization. In total, the 500 unseen concepts had 639,906 training examples that served as the unlabeled pool. We used 50 concepts for validation, leaving the remaining 450 concepts for our final experiments. The number of examples for each concept varied slightly, ranging from 0.114-0.203% of $|U|$. The 50,000 validation images were used as the test set.

**OpenImages** (Kuznetsova et al., 2020) has 7.34 million images with human-verified labels spread over 19,958 classes, taken as an unbiased sample from Flickr. However, only 6.82 million images were still available in the training set at the time of writing. As a feature extractor, we took ResNet-50 pre-trained on all of ImageNet and used the $l^2$ normalized output from the bottleneck layer. As rare concepts, we randomly selected 200 classes with between 100 to 6,817 positive training examples. We reviewed the selected classes and removed 47 classes that overlapped with ImageNet. The remaining 153 classes appeared in 0.002-0.088% of the data. We used the same hyperparameters as the ImageNet experiments and the OpenImages predefined test split for evaluation.

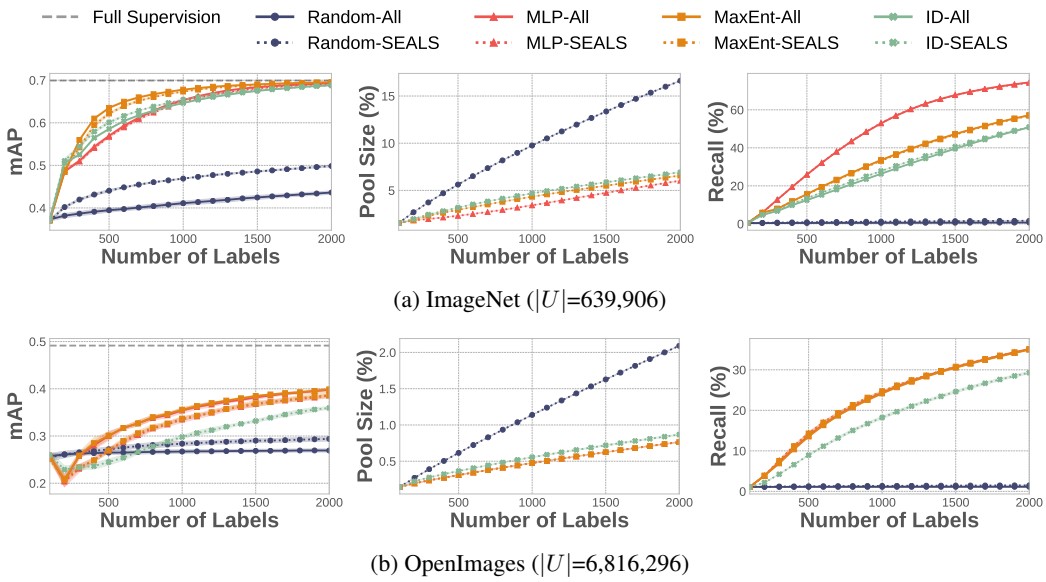

Figure 1: Active learning and search on ImageNet (top) and OpenImages (bottom). Across datasets and strategies, SEALS with $k = 100$ performed similarly to the baseline approach in terms of both the error the model achieved for active learning (left) and the recall of positive examples for active search (right), while only considering a fraction of the data $U$ (middle).

**10 billion (10B) images** from a large internet company were used to test SEALS' scalability. For the feature extractor, we used the same pre-trained ResNet-50 model as the OpenImages experiments. We also selected 8 additional classes from OpenImages as rare concepts: rat, sushi, bowling, beach, hawk, cupcake, and crowd. This allowed us to use the predefined test split from OpenImages for evaluation. Unlike the other datasets, we hired annotators to label images as they were selected and used a proprietary index to achieve low latency retrieval times to capture a real-world setting.

## 4.2 ACTIVE LEARNING

Across datasets and selection strategies, SEALS performed similarly to the baseline approaches that considered all of the unlabeled data in the candidate pool, as shown in Figures 1 and 2.

**ImageNet.** With a labeling budget of 2,000 examples per concept (~0.31% of $|U|$), all baseline and SEALS approaches ($k = 100$) were within 0.011 mAP of the 0.699 mAP achieved with full supervision. In contrast, random sampling (Random-All) only achieved 0.436 mAP. MLP-All, MaxEnt-All, and ID-All achieved mAPs of 0.693, 0.695, and 0.688, respectively, while the SEALS equivalents were all within 0.001 mAP at 0.692, 0.695, and 0.688 respectively and considered less than 7% of the unlabeled data. The resulting selection runtime for MLP-SEALS and MaxEnt-SEALS dropped by over 25×, leading to a 3.6× speed-up overall (Table 2). The speed-up was even larger for ID-SEALS, ranging from about 45× at 2,00 labels to 3000× at 200 labels. Even at a per-class level, the results were highly correlated with Pearson correlation coefficients of 0.9998 or more (Figure 10a in the Appendix). The reduced skew from the nearest neighbor expansion of the initial seed set only accounted for a small part of the improvement, as Random-SEALS achieved an mAP of 0.498.

**OpenImages.** The gap between the baseline approaches and SEALS widened slightly for OpenImages. At 2,000 labels per concept (~0.029% of $|U|$), MaxEnt-All and MLP-All achieved 0.399 and 0.398 mAP, respectively, while MaxEnt-SEALS and MLP-SEALS both achieved 0.386 mAP and considered less than 1% of the data. This sped-up the selection time by over 180× and the total time by over 3×. Increasing $k$ to 1,000 significantly narrowed this gap for MaxEnt-SEALS and MLP-SEALS, improving mAP to 0.395, as shown in the Appendix (Figure 7). Moreover, SEALS made ID tractable on OpenImages by reducing the candidate pool to 1% of the unlabeled data, whereas ID-All ran for over 24 hours in wall-clock time without completing a single round (Table 2).

Table 2: Wall clock runtimes for varying selection strategies on ImageNet and OpenImages. The last 3 columns break the total time down into 1) the time to apply the selection strategy to the candidate pool, 2) the time to find the $k$ nearest neighbors ($k$-NN) for the newly labeled examples, and 3) the time to train logistic regression on the currently labeled examples. Despite using a simple LSH index for similarity search, SEALS substantially improved runtimes across datasets and strategies.

| Dataset | Budget $T$ | Strategy $\phi$ | mAP/AUC | Recall (%) | Pool Size (%) | Total Time (seconds) | Time Breakdown (seconds) | | |
|---|---|---|---|---|---|---|---|---|---|
| | | | | | | | Selection | $k$-NN | Training |
| ImageNet | 2,000 | MaxEnt-All | 0.695 | 57.2 | 100.0 | 45.23 | 44.65 | - | 0.59 |
| | | MaxEnt-SEALS | 0.695 | 56.9 | 6.6 | 12.49 | 1.73 | 10.27 | 0.50 |
| | | MLP-All | 0.693 | 74.5 | 100.0 | 43.32 | 42.75 | - | 0.57 |
| | | MLP-SEALS | 0.692 | 74.2 | 6.0 | 12.03 | 1.48 | 9.94 | 0.63 |
| | | ID-All | 0.688 | 50.8 | 100.0 | 4654.59 | 4653.55 | - | 1.05 |
| | | ID-SEALS | 0.688 | 50.9 | 6.9 | 104.57 | 94.22 | 9.76 | 0.60 |
| | 1,000 | ID-All | 0.646 | 26.3 | 100.0 | 4620.04 | 4619.78 | - | 0.28 |
| | | ID-SEALS | 0.654 | 27.8 | 4.7 | 36.66 | 31.95 | 4.56 | 0.17 |
| | 500 | ID-All | 0.586 | 12.5 | 100.0 | 4602.64 | 4602.57 | - | 0.09 |
| | | ID-SEALS | 0.601 | 13.5 | 3.2 | 9.75 | 7.75 | 1.95 | 0.05 |
| | 200 | ID-All | 0.506 | 4.7 | 100.0 | 4588.76 | 4588.73 | - | 0.04 |
| | | ID-SEALS | 0.511 | 4.8 | 2.0 | 1.53 | 1.03 | 0.49 | 0.02 |
| OpenImages | 2,000 | MaxEnt-All | 0.399 | 35.0 | 100.0 | 295.20 | 294.78 | - | 0.42 |
| | | MaxEnt-SEALS | 0.386 | 35.1 | 0.8 | 80.61 | 1.56 | 78.63 | 0.43 |
| | | MLP-All | 0.398 | 35.1 | 100.0 | 285.27 | 284.88 | - | 0.40 |
| | | MLP-SEALS | 0.386 | 35.1 | 0.8 | 82.18 | 1.48 | 80.27 | 0.44 |
| | | ID-All | - | - | 100.0 | >24 hours | >24 hours | - | - |
| | | ID-SEALS | 0.359 | 29.3 | 0.9 | 129.79 | 48.98 | 80.40 | 0.41 |

**10B images.** Despite the unprecedented scale and limited pool size, SEALS performed similarly to the baseline approaches that scanned all 10 billion images. At a budget of 1,500 labels, MaxEnt-SEALS ($k$=10K) achieved a similar mAP to the baseline (0.504 vs. 0.508 mAP), while considering only about 0.1% of the data. This reduction allowed MaxEnt-SEALS to finish selection rounds in seconds on a single 24-core machine, while MaxEnt-All took minutes on a cluster with tens of thousands of cores. MLP-SEALS performed poorly at this scale because, for any image, there are likely many redundant or near-duplicate examples that provide little additional value.

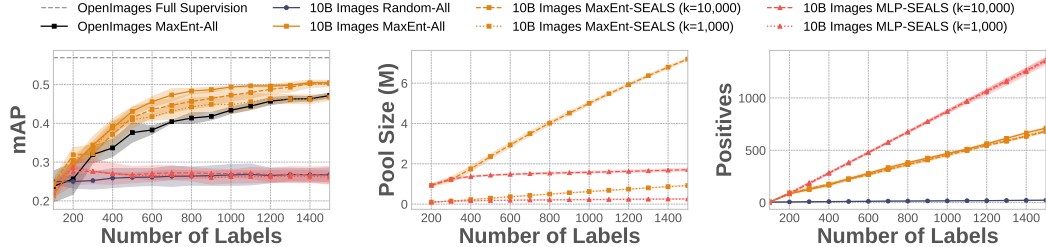

Figure 2: Active learning and search on a proprietary dataset of 10 billion images. Across strategies, SEALS with $k = 10,000$ performed similarly to the baseline approach in terms of both the error the model achieved for active learning (left) and the recall of positive examples for active search (right), while only considering a fraction of the data $U$ (middle).

### 4.3 ACTIVE SEARCH

As shown in Figures 1 and 2, SEALS recalled nearly the same number of positive examples as the baseline approaches did for all of the considered concepts, datasets, and selection strategies.

**ImageNet.** Unsurprisingly, MLP-All and MLP-SEALS significantly outperformed all of the other selection strategies for active search. At 2,000 labeled examples per concept, both approaches recalled over 74% of the positive examples for each concept at 74.5% and 74.2% recall, respectively. MaxEnt-All and MaxEnt-SEALS had a similar gap of 0.3%, labeling 57.2% and 56.9% of positive examples, while ID-All and ID-SEALS were even closer with a gap of only 0.1% (50.8% vs. 50.9%). Nearly all of the gains in recall are due to the selection strategies rather than the reduced skew in the initial seed, as Random-SEALS increased the recall by less than 1.0% over Random-All.

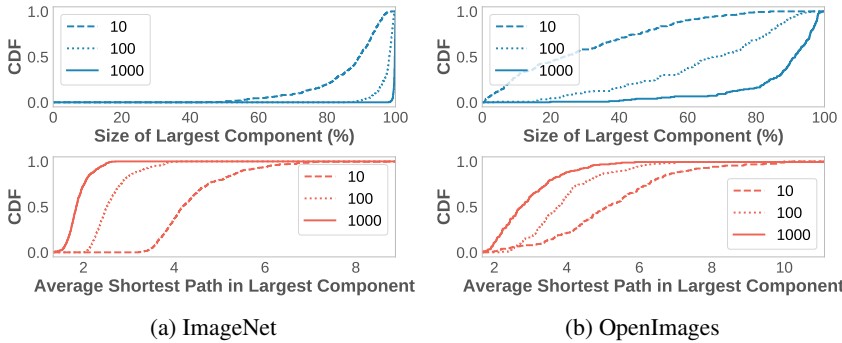

(a) ImageNet                                        (b) OpenImages

Figure 3: Measurements of the latent structure of unseen concepts in ImageNet (left) and OpenImages (right). Across datasets, the $k$-nearest neighbor graph of unseen concepts was well connected, forming large connected components (top) for even moderate values of $k$. The components were tightly packed, leading to short paths between examples (bottom).

**OpenImages.** The gap between the baseline approaches and SEALS was even closer on OpenImages despite considering a much smaller fraction of the overall unlabeled pool. MLP-All, MLP-SEALS, MaxEnt-SEALS, and MaxEnt-All were all within 0.1% with ~35% recall at 2,000 labels per concept. ID-SEALS had a recall of 29.3% but scaled nearly as well as the linear approaches.

**10B images.** SEALS performed as well as the baseline approach despite considering less than 0.1% of the data and collected 2 orders of magnitude more positive examples than random sampling.

### 4.4 LATENT STRUCTURE OF UNSEEN CONCEPTS

To better understand why and when SEALS works, we analyzed the nearest neighbor graph across concepts and values of $k$. Figure 3 shows the cumulative distribution functions (CDF) for the largest connected component within each concept and the average shortest paths between examples in that component. The 10B images dataset was excluded because only a few thousand examples were labeled. The largest connected component gives a sense of how much of the concept SEALS can reach, while the average shortest path serves as a proxy for how long it will take to explore.

In general, SEALS performed better for concepts that formed larger connected components and had shorter paths between examples (Figure 11 in the Appendix). For most concepts in ImageNet, the largest connected component contained the majority of examples, and the paths between examples were very short. These tight clusters explain why so few examples were needed to learn accurate binary concept classifiers, as shown in Section 4.2, and why SEALS recovered ~74% of positive examples on average while only labeling ~0.31% of the data. If we constructed the candidate pool by randomly selecting examples, mAP and recall would drop for all strategies (Appendix A.5). The concepts were so rare that the randomly chosen examples were not close to the decision boundary. For OpenImages, rare concepts were more fragmented, but each component was fairly tight, leading to short paths between examples. On a per-class level, concepts like "monster truck" and "blackberry" performed much better than generic concepts like "electric blue" and "meal" that were more scattered (Appendix A.6 and A.7). This fragmentation partly explains the gap between SEALS and the baselines in Section 4.2, and why increasing $k$ closed it. However, even for small values of $k$, there were significant gains over random sampling, as shown in Figures 6 and 7 in the Appendix.

## 5 CONCLUSION

In this work, we introduced Similarity search for Efficient Active Learning and Search (SEALS) as a simple approach to accelerate active learning and search that can be applied to a wide range of existing algorithms. SEALS restricted the candidate pool for labeling to the nearest neighbors of the currently labeled set instead of scanning over all of the unlabeled data. Across three large datasets, three selection strategies, and 611 concepts, we found that SEALS achieved similar model quality and recall of positive examples while improving computational efficiency by orders of magnitude.

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

# A APPENDIX

## A.1 NUMBER OF INITIAL POSITIVES

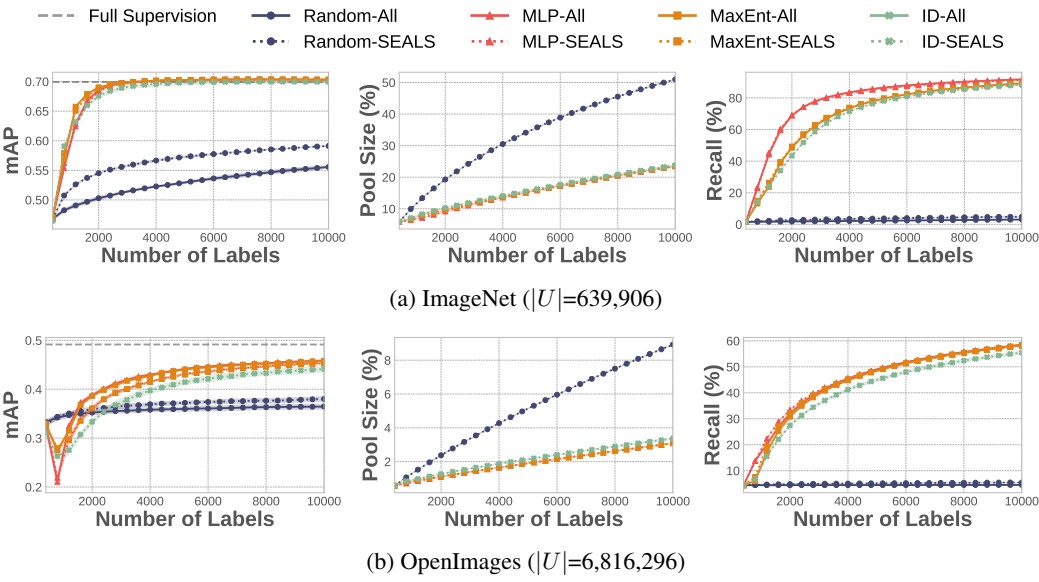

(a) ImageNet ($|U|$=639,906)

(b) OpenImages ($|U|$=6,816,296)

Figure 4: **Active learning and search with 20 positive seed examples** and a labeling budget of 10,000 examples on ImageNet (top) and OpenImages (bottom). Across datasets and strategies, SEALS with $k = 100$ performs similarly to the baseline approach in terms of both the error the model achieves for active learning (left) and the recall of positive examples for active search (right), while only considering a fraction of the unlabeled data $U$ (middle).

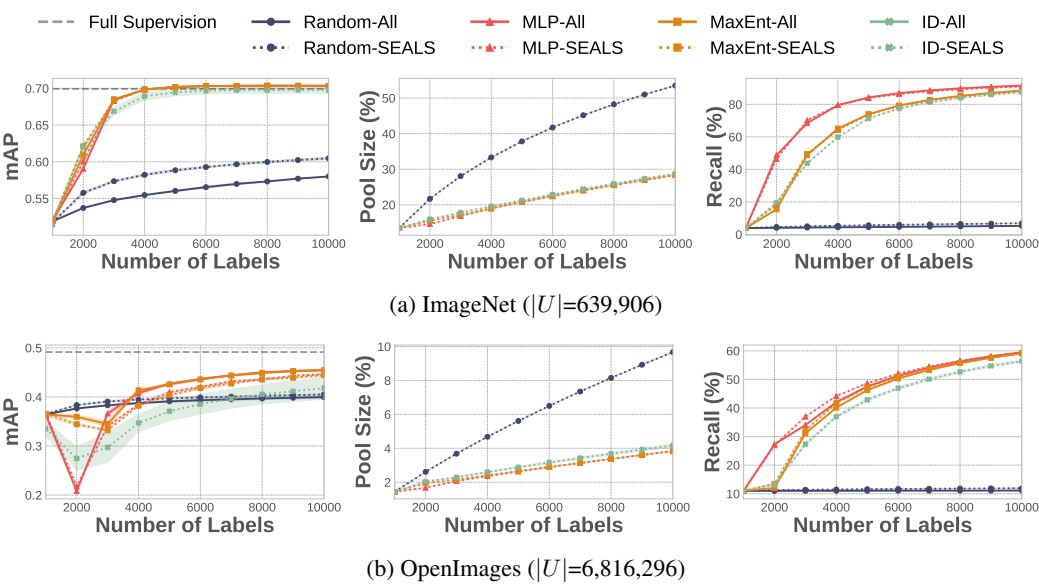

(a) ImageNet ($|U|$=639,906)

(b) OpenImages ($|U|$=6,816,296)

Figure 5: **Active learning and search with 50 positive seed examples** and a labeling budget of 10,000 examples on ImageNet (top) and OpenImages (bottom). Across datasets and strategies, SEALS with $k = 100$ performs similarly to the baseline approach in terms of both the error the model achieves for active learning (left) and the recall of positive examples for active search (right), while only considering a fraction of the unlabeled data $U$ (middle).

## A.2 IMPACT OF $k$ ON SEALS

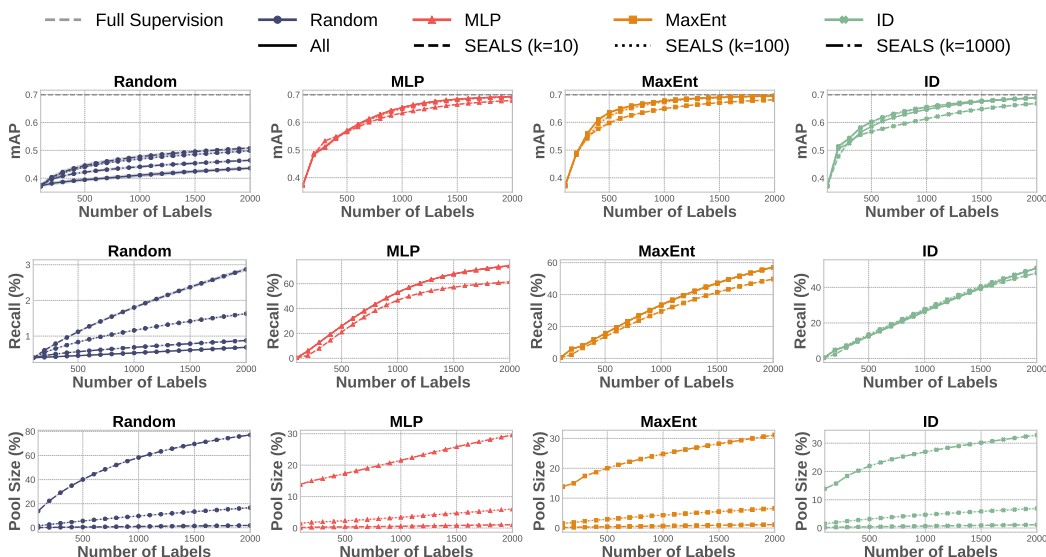

Figure 6: **Impact of increasing $k$ on ImageNet ($|U|$=639,906).** Larger values of $k$ help to close the gap between SEALS and the baseline approach that considers all of the unlabeled data for both active learning (top) and active search (middle). However, increasing $k$ also increases the candidate pool size (bottom), presenting a trade-off between labeling efficiency and computational efficiency.

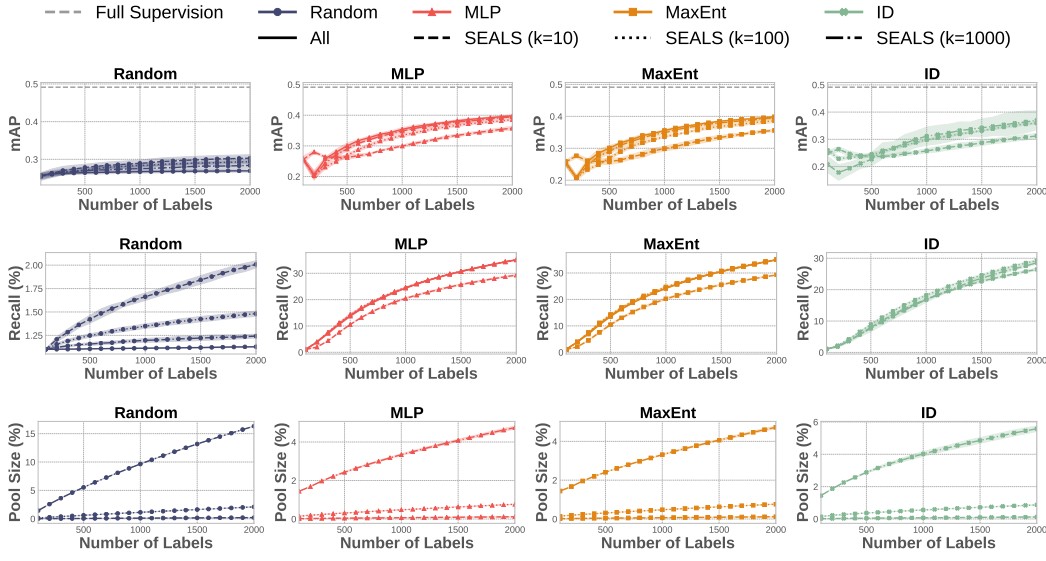

Figure 7: **Impact of increasing $k$ on OpenImages ($|U|$=6,816,296).** Larger values of $k$ help to close the gap between SEALS and the baseline approach that considers all of the unlabeled data for both active learning (top) and active search (middle). However, increasing $k$ also increases the candidate pool size (bottom), presenting a trade-off between labeling efficiency and computational efficiency.

## A.3 SELF-SUPERVISED EMBEDDINGS (SIMCLR) ON IMAGENET

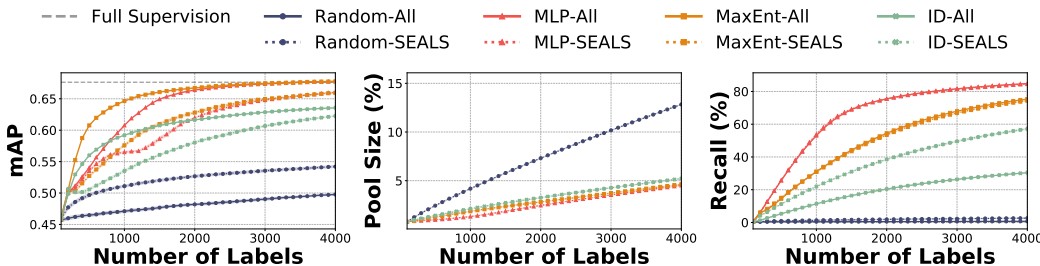

Figure 8: Active learning and search on ImageNet with self-supervised embeddings from Sim-CLR (Chen et al., 2020). Because the self-supervised training for the embeddings did not use the labels, results are average across all 1,000 classes and $|U|$=1,281,167. To compensate for the larger unlabeled pool, we extended the total labeling budget to 4,000 compared to the 2,000 used in Figure 1. Across strategies, SEALS with $k = 100$ substantially outperforms random sampling in terms of both the mAP the model achieves for active learning (left) and the recall of positive examples for active search (right), while only considering a fraction of the data $U$ (middle). For active learning, the gap between the baseline and SEALS approaches is slightly larger than in Figure 1, which is likely due to the larger pool size and increased average shortest paths (see Figure 9).

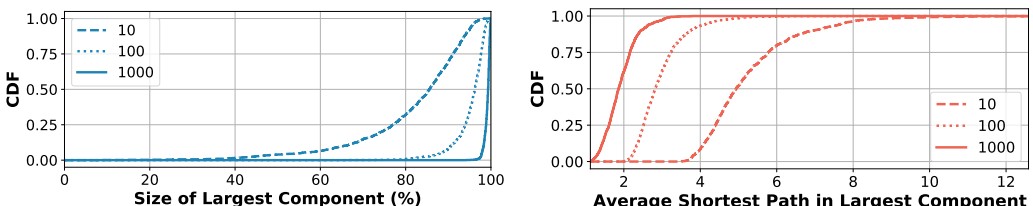

Figure 9: Measurements of the latent structure of unseen concepts in ImageNet with self-supervised embeddings from SimCLR (Chen et al., 2020). In comparison to Figure 3a, the $k$-nearest neighbor graph for unseen concepts was still well connected, forming large connected components (left) for even moderate values of $k$, but the average shortest path between examples was slightly longer (right). The increased path length is not too surprising considering the fully supervised model still outperformed the linear evaluation of the self-supervised embeddings in Chen et al. (2020).

## A.4 Per class AP correlation

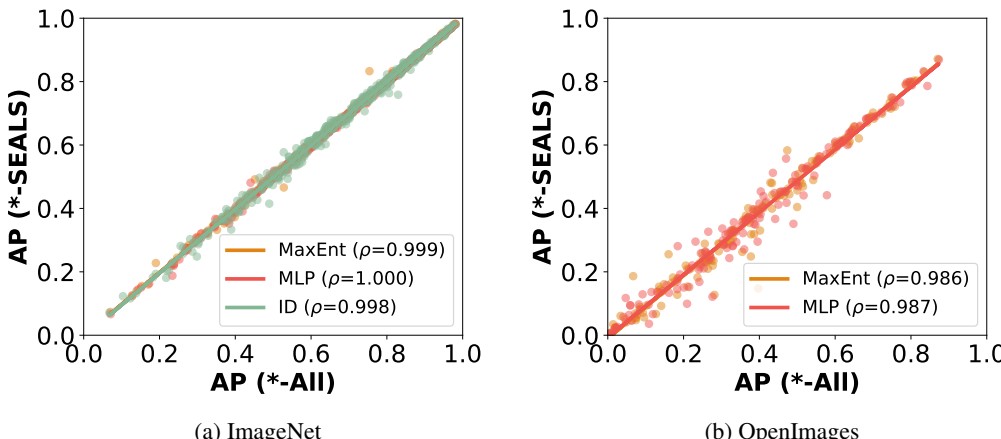

(a) ImageNet

(b) OpenImages

Figure 10: The per-class APs of SEALS were highly correlated to the baseline approaches (*-All) for active learning on ImageNet (right) and OpenImages (left). On OpenImages with $k = 100$ and a budget of 2,000 labels, the Pearson's correlation ($\rho$) between the baseline and SEALS for the average precision of individual classes was 0.986 for MaxEnt and 0.987 for MLP. The least-squares fit had a slope of 0.99 and y-intercept of -0.01. On ImageNet, the correlations were even higher.

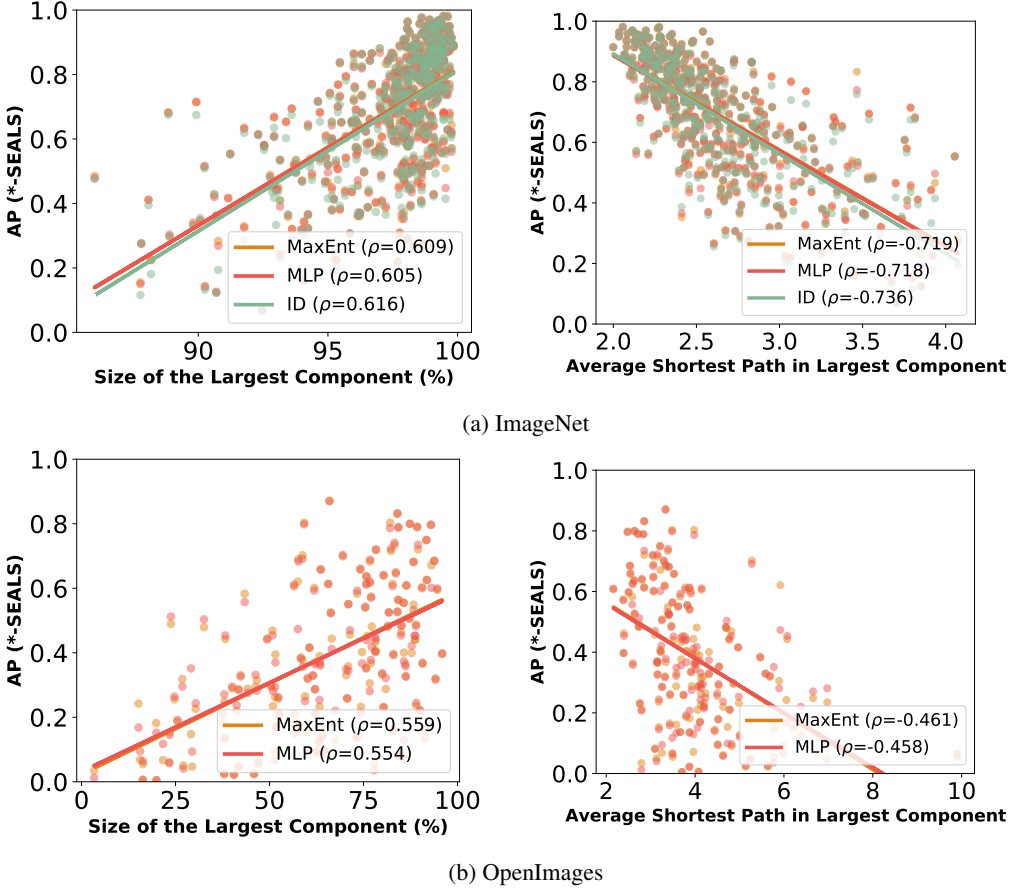

(a) ImageNet

(b) OpenImages

Figure 11: SEALS achieved higher APs for classes that formed larger connected components (left) and had shorter paths between examples (right) in ImageNet (top) and OpenImages (bottom).

## A.5 COMPARISON TO POOL OF RANDOMLY SELECTED EXAMPLES

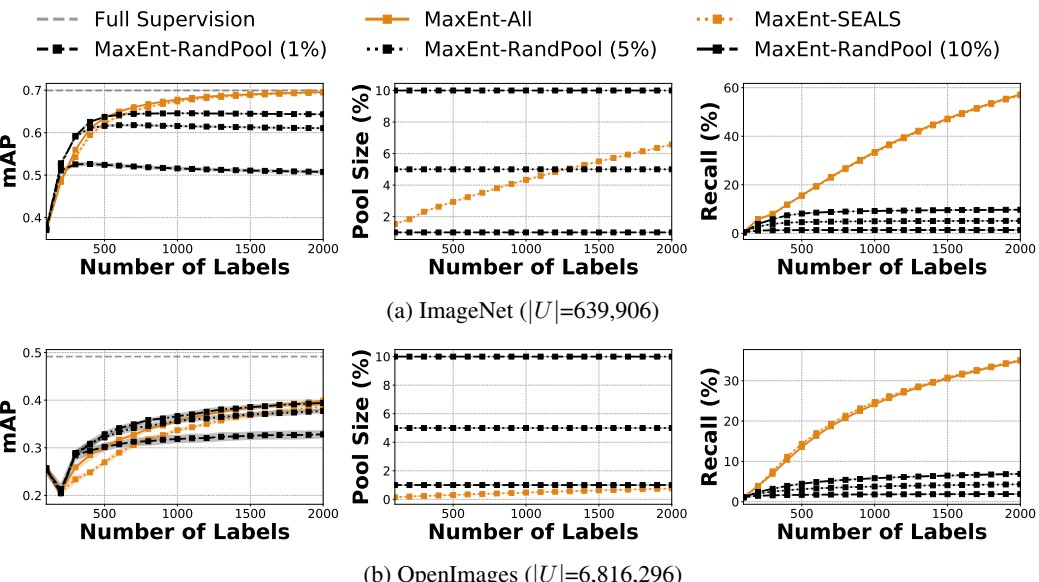

(a) ImageNet ($|U|$=639,906)

(b) OpenImages ($|U|$=6,816,296)

Figure 12: MaxEnt-SEALS ($k = 100$) versus MaxEnt applied to a candidate pool of randomly selected examples (RandPool). Because the concepts we considered were so rare, as is often the case in practice, randomly chosen examples are unlikely to be close to the decision boundary, and a much larger pool is required to match SEALS. On ImageNet (top), MaxEnt-SEALS outperformed MaxEnt-RandPool in terms of both the error the model achieves for active learning (left) and the recall of positive examples for active search (right) even with a pool containing 10% of the data (middle). On Openimages (bottom), MaxEnt-RandPool needed at least $5\times$ as much data to match MaxEnt-SEALS for active learning and failed to achieve similar recall even with $10\times$ the data.

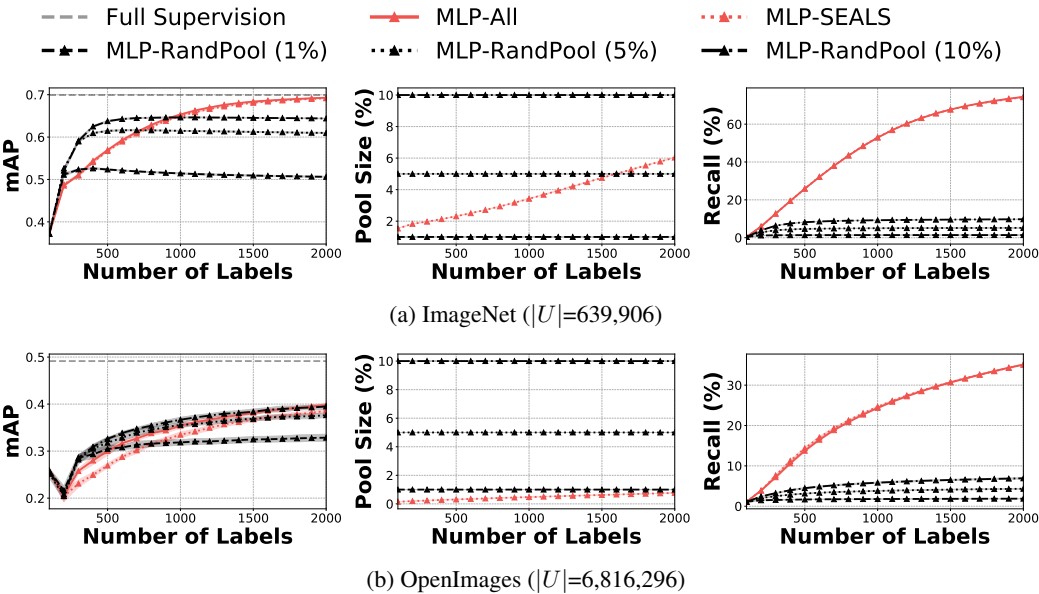

(a) ImageNet ($|U|$=639,906)

(b) OpenImages ($|U|$=6,816,296)

Figure 13: MLP-SEALS ($k = 100$) versus MLP applied to a candidate pool of randomly selected examples (RandPool). Because the concepts we considered were so rare, as is often the case in practice, randomly chosen examples are unlikely to be close to the decision boundary, and a much larger pool is required to match SEALS. On ImageNet (top), MLP-SEALS outperformed MLP-RandPool in terms of both the error the model achieves for active learning (left) and the recall of positive examples for active search (right) even with a pool containing 10% of the data (middle). On Openimages (bottom), MLP-RandPool needed at least $5\times$ as much data to match MLP-SEALS for active learning and failed to achieve similar recall even with $10\times$ the data.

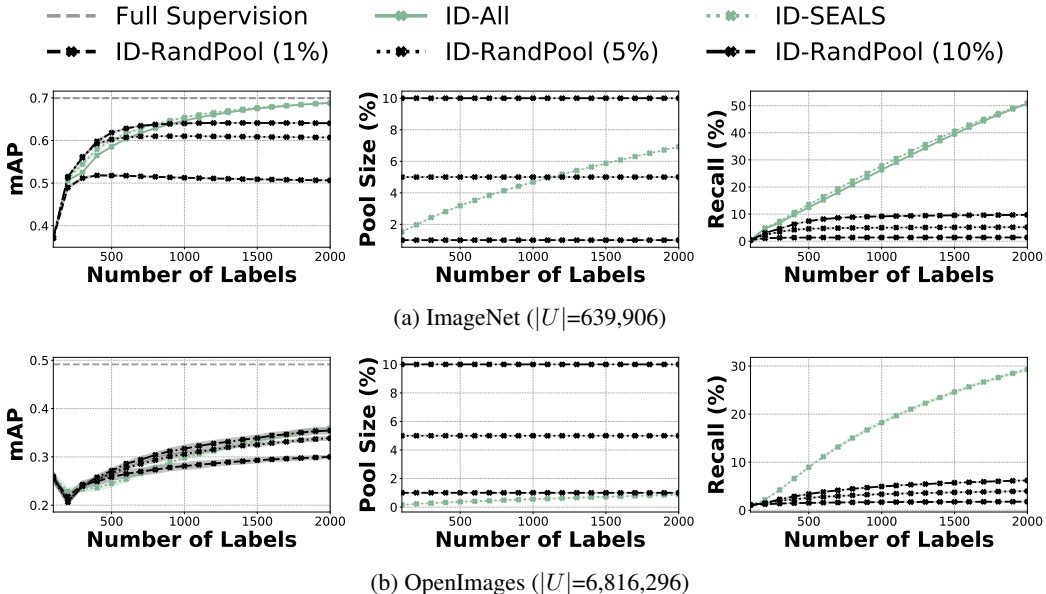

(a) ImageNet ($|U|$=639,906)

(b) OpenImages ($|U|$=6,816,296)

Figure 14: ID-SEALS ($k = 100$) versus ID applied to a candidate pool of randomly selected examples (RandPool). Because the concepts we considered were so rare, as is often the case in practice, randomly chosen examples are unlikely to be close to the decision boundary, and a much larger pool is required to match SEALS. On ImageNet (top), ID-SEALS outperformed ID-RandPool in terms of both the error the model achieves for active learning (left) and the recall of positive examples for active search (right) even with a pool containing 10% of the data (middle). On Openimages (bottom), ID-RandPool needed at least $5\times$ as much data to match ID-SEALS for active learning and failed to achieve similar recall even with $10\times$ the data.

A.6   ACTIVE LEARNING ON EACH SELECTED CLASS FROM OPENIMAGES

Table 3: **Top $\frac{1}{3}$ of classes from Openimages for active learning.** (1 of 3) Average precision and measurements of the largest component (LC) for each selected class (153 total) from OpenImages with a labeling budget of 2,000 examples. Classes are ordered based on MaxEnt-SEALS.

| Display Name | Total Positives | Size of the LC (%) | Average Shortest Path in the LC | Random (All) | MaxEnt (SEALS) | MaxEnt (All) | Full Supervision |
|---|---|---|---|---|---|---|---|
| Citrus | 796 | 65 | 3.34 | 0.34 | 0.87 | 0.87 | 0.87 |
| Cargo ship | 219 | 84 | 2.85 | 0.70 | 0.83 | 0.83 | 0.86 |
| Blackberry | 245 | 87 | 2.64 | 0.67 | 0.80 | 0.80 | 0.79 |
| Galliformes | 674 | 82 | 3.98 | 0.72 | 0.80 | 0.82 | 0.92 |
| Rope | 618 | 59 | 3.48 | 0.29 | 0.80 | 0.81 | 0.74 |
| Hurdling | 269 | 92 | 2.48 | 0.26 | 0.80 | 0.79 | 0.80 |
| Roman temple | 345 | 89 | 2.72 | 0.63 | 0.79 | 0.79 | 0.82 |
| Monster truck | 286 | 84 | 2.84 | 0.41 | 0.79 | 0.80 | 0.81 |
| Pasta | 954 | 91 | 3.21 | 0.42 | 0.75 | 0.75 | 0.79 |
| Chess | 740 | 83 | 3.39 | 0.53 | 0.73 | 0.74 | 0.86 |
| Bowed string instrument | 728 | 78 | 3.05 | 0.72 | 0.72 | 0.74 | 0.79 |
| Parrot | 1546 | 89 | 2.85 | 0.59 | 0.72 | 0.76 | 0.92 |
| Calabaza | 870 | 82 | 3.15 | 0.50 | 0.71 | 0.75 | 0.81 |
| Superhero | 968 | 58 | 5.28 | 0.17 | 0.70 | 0.70 | 0.67 |
| Drums | 741 | 69 | 3.30 | 0.52 | 0.70 | 0.72 | 0.83 |
| Shooting range | 189 | 57 | 3.06 | 0.38 | 0.69 | 0.69 | 0.68 |
| Ancient roman architecture | 589 | 76 | 3.34 | 0.61 | 0.68 | 0.70 | 0.77 |
| Cupboard | 898 | 88 | 3.41 | 0.53 | 0.68 | 0.69 | 0.75 |
| Ibis | 259 | 93 | 2.53 | 0.29 | 0.68 | 0.69 | 0.66 |
| Cattle | 5995 | 93 | 3.22 | 0.37 | 0.67 | 0.68 | 0.74 |
| Galleon | 182 | 74 | 2.54 | 0.45 | 0.66 | 0.66 | 0.61 |
| Kitchen knife | 360 | 63 | 3.52 | 0.32 | 0.66 | 0.65 | 0.66 |
| Grapefruit | 506 | 83 | 3.06 | 0.50 | 0.65 | 0.65 | 0.69 |
| Deacon | 341 | 80 | 2.80 | 0.48 | 0.64 | 0.64 | 0.67 |
| Rye | 128 | 75 | 2.63 | 0.51 | 0.64 | 0.64 | 0.65 |
| Chartreux | 147 | 91 | 2.59 | 0.50 | 0.63 | 0.63 | 0.69 |
| San Pedro cactus | 318 | 76 | 3.32 | 0.17 | 0.62 | 0.63 | 0.71 |
| Skateboarding Equipment | 862 | 57 | 5.92 | 0.20 | 0.62 | 0.66 | 0.66 |
| Electric piano | 345 | 56 | 4.15 | 0.24 | 0.61 | 0.60 | 0.48 |
| Straw | 547 | 65 | 2.85 | 0.33 | 0.61 | 0.62 | 0.61 |
| Berry | 874 | 82 | 3.78 | 0.30 | 0.61 | 0.61 | 0.69 |
| East-european shepherd | 206 | 86 | 2.16 | 0.61 | 0.61 | 0.62 | 0.65 |
| Ring | 676 | 75 | 3.87 | 0.15 | 0.61 | 0.64 | 0.64 |
| Rat | 1151 | 94 | 2.50 | 0.32 | 0.60 | 0.60 | 0.61 |
| Coral reef fish | 434 | 90 | 3.07 | 0.51 | 0.60 | 0.64 | 0.79 |
| Concert dance | 357 | 61 | 3.91 | 0.37 | 0.60 | 0.60 | 0.70 |
| Whole food | 708 | 73 | 3.66 | 0.18 | 0.58 | 0.60 | 0.57 |
| Modern pentathlon | 772 | 43 | 2.59 | 0.13 | 0.58 | 0.47 | 0.51 |
| Gymnast | 235 | 77 | 2.39 | 0.39 | 0.57 | 0.59 | 0.65 |
| California roll | 368 | 84 | 3.49 | 0.05 | 0.56 | 0.56 | 0.58 |
| Shrimp | 907 | 85 | 3.82 | 0.07 | 0.56 | 0.56 | 0.58 |
| Log cabin | 448 | 70 | 3.62 | 0.44 | 0.55 | 0.55 | 0.62 |
| Formula racing | 351 | 88 | 3.38 | 0.33 | 0.55 | 0.54 | 0.60 |
| Herd | 648 | 75 | 3.88 | 0.42 | 0.54 | 0.55 | 0.67 |
| Embroidery | 356 | 81 | 3.41 | 0.32 | 0.53 | 0.53 | 0.60 |
| Shelving | 810 | 66 | 3.41 | 0.27 | 0.53 | 0.53 | 0.51 |
| Downhill | 194 | 84 | 2.64 | 0.42 | 0.53 | 0.51 | 0.59 |
| Daylily | 391 | 87 | 3.25 | 0.20 | 0.51 | 0.50 | 0.49 |
| Automotive exterior | 1060 | 23 | 2.74 | 0.65 | 0.49 | 0.54 | 0.69 |
| Ciconiiformes | 426 | 88 | 3.47 | 0.33 | 0.49 | 0.51 | 0.48 |
| Monoplane | 756 | 81 | 4.70 | 0.13 | 0.48 | 0.43 | 0.48 |

Table 4: **Middle $\frac{1}{3}$ of classes from Openimages for active learning.** (2 of 3) Average precision and measurements of the largest component (LC) for each selected class (153 total) from OpenImages with a labeling budget of 2,000 examples. Classes are ordered based on MaxEnt-SEALS.

| Display Name | Total Positives | Size of the LC (%) | Average Shortest Path in the LC | Random (All) | MaxEnt (SEALS) | MaxEnt (All) | Full Supervision |
|---|---|---|---|---|---|---|---|
| Seafood boil | 322 | 85 | 2.73 | 0.31 | 0.48 | 0.49 | 0.51 |
| Landscaping | 789 | 32 | 4.71 | 0.26 | 0.48 | 0.51 | 0.63 |
| Skating | 561 | 77 | 4.04 | 0.17 | 0.48 | 0.43 | 0.40 |
| Floodplain | 567 | 50 | 4.81 | 0.61 | 0.47 | 0.52 | 0.66 |
| Knitting | 409 | 71 | 3.10 | 0.61 | 0.46 | 0.50 | 0.73 |
| Elk | 353 | 84 | 2.40 | 0.15 | 0.46 | 0.48 | 0.45 |
| Bilberry | 228 | 75 | 3.77 | 0.10 | 0.45 | 0.45 | 0.32 |
| Goat | 1190 | 88 | 3.72 | 0.17 | 0.44 | 0.45 | 0.61 |
| Fortification | 287 | 66 | 3.96 | 0.43 | 0.44 | 0.46 | 0.52 |
| Annual plant | 677 | 38 | 6.07 | 0.39 | 0.44 | 0.43 | 0.58 |
| Mcdonnell douglas f/a-18 hornet | 160 | 88 | 3.51 | 0.11 | 0.44 | 0.47 | 0.37 |
| Tooth | 976 | 49 | 4.77 | 0.16 | 0.44 | 0.48 | 0.56 |
| Briefs | 539 | 78 | 3.68 | 0.15 | 0.43 | 0.44 | 0.46 |
| Sirloin steak | 297 | 60 | 4.97 | 0.14 | 0.42 | 0.42 | 0.46 |
| Smoothie | 330 | 78 | 3.22 | 0.15 | 0.41 | 0.41 | 0.38 |
| Glider | 393 | 82 | 3.94 | 0.08 | 0.40 | 0.40 | 0.48 |
| Bathroom cabinet | 368 | 95 | 2.39 | 0.29 | 0.40 | 0.39 | 0.37 |
| White-tailed deer | 238 | 87 | 3.24 | 0.34 | 0.40 | 0.43 | 0.43 |
| Bird of prey | 712 | 78 | 3.81 | 0.76 | 0.40 | 0.50 | 0.91 |
| Egg (Food) | 1193 | 85 | 4.31 | 0.14 | 0.40 | 0.37 | 0.63 |
| Soldier | 1032 | 74 | 3.80 | 0.62 | 0.40 | 0.41 | 0.72 |
| Cranberry | 450 | 63 | 4.10 | 0.13 | 0.39 | 0.39 | 0.37 |
| Estate | 667 | 51 | 4.03 | 0.47 | 0.39 | 0.40 | 0.54 |
| Chocolate truffle | 288 | 58 | 5.47 | 0.10 | 0.39 | 0.40 | 0.42 |
| Town square | 617 | 58 | 3.69 | 0.31 | 0.38 | 0.36 | 0.47 |
| Bakmi | 191 | 76 | 3.34 | 0.27 | 0.37 | 0.37 | 0.36 |
| Trail riding | 679 | 90 | 3.15 | 0.21 | 0.37 | 0.37 | 0.38 |
| Aerial photography | 931 | 63 | 3.99 | 0.39 | 0.37 | 0.37 | 0.66 |
| Lugger | 103 | 62 | 3.14 | 0.35 | 0.37 | 0.37 | 0.42 |
| Paddy field | 468 | 70 | 4.02 | 0.17 | 0.36 | 0.36 | 0.43 |
| Pavlova | 195 | 86 | 2.60 | 0.19 | 0.36 | 0.36 | 0.34 |
| Steamed rice | 580 | 75 | 4.54 | 0.10 | 0.35 | 0.37 | 0.48 |
| Pancit | 385 | 86 | 3.16 | 0.21 | 0.33 | 0.33 | 0.31 |
| Factory | 333 | 61 | 5.59 | 0.17 | 0.33 | 0.34 | 0.35 |
| Fur | 834 | 42 | 4.31 | 0.08 | 0.33 | 0.33 | 0.31 |
| Stallion | 598 | 70 | 3.58 | 0.32 | 0.33 | 0.40 | 0.64 |
| Optical instrument | 649 | 79 | 3.91 | 0.15 | 0.33 | 0.33 | 0.28 |
| Thumb | 895 | 26 | 4.18 | 0.07 | 0.32 | 0.39 | 0.41 |
| Meal | 1250 | 60 | 5.68 | 0.52 | 0.32 | 0.38 | 0.59 |
| American shorthair | 2084 | 94 | 3.32 | 0.12 | 0.32 | 0.32 | 0.24 |
| Bracelet | 770 | 46 | 4.13 | 0.09 | 0.31 | 0.33 | 0.24 |
| Vehicle registration plate | 5697 | 76 | 5.89 | 0.28 | 0.31 | 0.33 | 0.53 |
| Ice | 682 | 50 | 4.87 | 0.23 | 0.30 | 0.32 | 0.55 |
| Lamian | 257 | 80 | 3.57 | 0.23 | 0.29 | 0.32 | 0.28 |
| Multimedia | 741 | 46 | 4.12 | 0.45 | 0.29 | 0.31 | 0.53 |
| Belt | 467 | 41 | 3.26 | 0.06 | 0.29 | 0.31 | 0.31 |
| Prairie | 792 | 44 | 3.92 | 0.37 | 0.29 | 0.26 | 0.57 |
| Boardsport | 673 | 62 | 4.08 | 0.26 | 0.29 | 0.29 | 0.53 |
| Asphalt | 1026 | 40 | 4.53 | 0.23 | 0.29 | 0.29 | 0.45 |
| Costume design | 818 | 52 | 3.44 | 0.07 | 0.26 | 0.26 | 0.28 |
| Cottage | 670 | 51 | 4.13 | 0.36 | 0.26 | 0.36 | 0.61 |

Table 5: **Bottom $\frac{1}{3}$ of classes from Openimages for active learning.** (3 of 3) Average precision and measurements of the largest component (LC) for each selected class (153 total) from OpenImages with a labeling budget of 2,000 examples. Classes are ordered based on MaxEnt-SEALS.

| Display Name | Total Positives | Size of the LC (%) | Average Shortest Path in the LC | Random (All) | MaxEnt (SEALS) | MaxEnt (All) | Full Supervision |
|---|---|---|---|---|---|---|---|
| Stele | 450 | 70 | 3.74 | 0.12 | 0.26 | 0.25 | 0.35 |
| Mode of transport | 1387 | 24 | 4.50 | 0.15 | 0.26 | 0.16 | 0.54 |
| Temperate coniferous forest | 328 | 59 | 4.23 | 0.30 | 0.26 | 0.29 | 0.40 |
| Bumper | 985 | 37 | 6.65 | 0.49 | 0.25 | 0.38 | 0.64 |
| Interaction | 924 | 15 | 6.05 | 0.04 | 0.24 | 0.25 | 0.37 |
| Plumbing fixture | 2124 | 89 | 3.19 | 0.31 | 0.24 | 0.27 | 0.38 |
| Shorebird | 234 | 80 | 2.76 | 0.32 | 0.23 | 0.26 | 0.37 |
| Icing | 1118 | 74 | 4.20 | 0.13 | 0.23 | 0.25 | 0.46 |
| Wilderness | 1225 | 30 | 4.12 | 0.29 | 0.23 | 0.24 | 0.39 |
| Construction | 515 | 63 | 4.99 | 0.13 | 0.23 | 0.26 | 0.34 |
| Carpet | 644 | 50 | 6.98 | 0.05 | 0.23 | 0.28 | 0.43 |
| Maple | 2301 | 90 | 4.19 | 0.06 | 0.22 | 0.21 | 0.36 |
| Rural area | 921 | 41 | 4.63 | 0.33 | 0.22 | 0.28 | 0.50 |
| Singer | 604 | 56 | 4.06 | 0.12 | 0.21 | 0.21 | 0.40 |
| Delicatessen | 196 | 52 | 2.80 | 0.14 | 0.21 | 0.22 | 0.27 |
| Canal | 726 | 62 | 4.78 | 0.22 | 0.21 | 0.26 | 0.46 |
| Organ (Biology) | 1156 | 25 | 3.80 | 0.23 | 0.19 | 0.07 | 0.44 |
| Laugh | 750 | 19 | 6.22 | 0.06 | 0.18 | 0.17 | 0.26 |
| Plateau | 452 | 37 | 3.88 | 0.41 | 0.18 | 0.24 | 0.46 |
| Algae | 426 | 57 | 4.52 | 0.15 | 0.18 | 0.19 | 0.26 |
| Cactus | 377 | 51 | 4.11 | 0.05 | 0.17 | 0.18 | 0.22 |
| Engine | 656 | 82 | 3.43 | 0.16 | 0.17 | 0.17 | 0.26 |
| Marine mammal | 2954 | 91 | 3.58 | 0.19 | 0.16 | 0.15 | 0.21 |
| Frost | 483 | 60 | 4.73 | 0.20 | 0.15 | 0.21 | 0.47 |
| Paper | 969 | 23 | 3.18 | 0.16 | 0.15 | 0.14 | 0.41 |
| Cirque | 347 | 29 | 5.77 | 0.43 | 0.15 | 0.40 | 0.55 |
| Pork | 464 | 64 | 4.44 | 0.06 | 0.14 | 0.14 | 0.15 |
| Antenna | 545 | 73 | 3.66 | 0.10 | 0.14 | 0.13 | 0.29 |
| Portrait | 2510 | 67 | 6.38 | 0.23 | 0.13 | 0.18 | 0.43 |
| Flooring | 814 | 38 | 3.87 | 0.10 | 0.13 | 0.14 | 0.20 |
| Cycling | 794 | 63 | 5.00 | 0.53 | 0.13 | 0.28 | 0.66 |
| Chevrolet silverado | 115 | 62 | 4.82 | 0.05 | 0.09 | 0.08 | 0.12 |
| Tool | 1549 | 64 | 4.51 | 0.08 | 0.09 | 0.10 | 0.13 |
| Liqueur | 539 | 51 | 5.98 | 0.26 | 0.09 | 0.14 | 0.38 |
| Pleurotus eryngii | 140 | 84 | 3.10 | 0.11 | 0.08 | 0.08 | 0.14 |
| Organism | 1148 | 21 | 3.49 | 0.05 | 0.07 | 0.13 | 0.26 |
| Pelecaniformes | 457 | 85 | 3.96 | 0.30 | 0.07 | 0.09 | 0.32 |
| Icon | 186 | 15 | 3.26 | 0.05 | 0.07 | 0.07 | 0.16 |
| Stadium | 1654 | 77 | 5.77 | 0.35 | 0.06 | 0.10 | 0.48 |
| Space | 1006 | 23 | 4.63 | 0.03 | 0.06 | 0.03 | 0.14 |
| Performing arts | 1030 | 29 | 6.97 | 0.12 | 0.05 | 0.06 | 0.53 |
| Mural | 649 | 41 | 5.24 | 0.13 | 0.05 | 0.07 | 0.34 |
| Brown | 1427 | 16 | 3.49 | 0.02 | 0.05 | 0.07 | 0.20 |
| Wall | 1218 | 27 | 3.13 | 0.11 | 0.05 | 0.05 | 0.27 |
| Tournament | 841 | 47 | 9.90 | 0.15 | 0.05 | 0.07 | 0.16 |
| White | 1494 | 3 | 2.79 | 0.02 | 0.03 | 0.01 | 0.10 |
| Mitsubishi | 511 | 37 | 5.14 | 0.01 | 0.02 | 0.02 | 0.04 |
| Exhibition | 513 | 40 | 3.87 | 0.03 | 0.02 | 0.02 | 0.14 |
| Scale model | 667 | 45 | 5.64 | 0.05 | 0.02 | 0.02 | 0.13 |
| Teal | 975 | 16 | 4.08 | 0.01 | 0.01 | 0.01 | 0.04 |
| Electric blue | 1180 | 19 | 3.70 | 0.01 | 0.00 | 0.01 | 0.06 |

## A.7 ACTIVE SEARCH ON EACH SELECTED CLASS FROM OPENIMAGES

Table 6: **Top $\frac{1}{3}$ of classes from Openimages for active search.** (1 of 3) Recall (%) of positives and measurements of the largest component (LC) for each selected class (153 total) from OpenImages with a labeling budget of 2,000 examples. Classes are ordered based on MLP-SEALS.

| Display Name | Total Positives | Size of the LC (%) | Average Shortest Path in the LC | Random (All) | MLP (SEALS) | MLP (All) |
|---|---|---|---|---|---|---|
| Chartreux | 147 | 91 | 2.59 | 3.5 | 83.9 | 84.6 |
| Ibis | 259 | 93 | 2.53 | 2.0 | 83.9 | 83.9 |
| Hurdling | 269 | 92 | 2.48 | 1.9 | 83.5 | 86.2 |
| East-european shepherd | 206 | 86 | 2.16 | 2.4 | 78.2 | 78.3 |
| Blackberry | 245 | 87 | 2.64 | 2.0 | 77.5 | 78.5 |
| Bathroom cabinet | 368 | 95 | 2.39 | 1.4 | 76.8 | 77.1 |
| Rat | 1151 | 94 | 2.50 | 0.5 | 75.1 | 75.2 |
| Rye | 128 | 75 | 2.63 | 3.9 | 74.7 | 74.5 |
| Elk | 353 | 84 | 2.40 | 1.5 | 73.4 | 74.3 |
| Pavlova | 195 | 86 | 2.60 | 2.6 | 70.8 | 71.3 |
| Seafood boil | 322 | 85 | 2.73 | 1.6 | 70.4 | 70.6 |
| Roman temple | 345 | 89 | 2.72 | 1.5 | 69.2 | 68.3 |
| Monster truck | 286 | 84 | 2.84 | 1.7 | 68.1 | 67.8 |
| Downhill | 194 | 84 | 2.64 | 2.6 | 67.2 | 69.0 |
| Shorebird | 234 | 80 | 2.76 | 2.1 | 66.8 | 66.4 |
| Mcdonnell douglas f/a-18 hornet | 160 | 88 | 3.51 | 3.2 | 66.0 | 67.9 |
| San Pedro cactus | 318 | 76 | 3.32 | 1.6 | 65.8 | 64.9 |
| Pleurotus eryngii | 140 | 84 | 3.10 | 3.6 | 65.7 | 66.1 |
| California roll | 368 | 84 | 3.49 | 1.4 | 65.3 | 68.0 |
| Gymnast | 235 | 77 | 2.39 | 2.2 | 64.0 | 64.0 |
| Galleon | 182 | 74 | 2.54 | 2.7 | 62.4 | 61.5 |
| Cargo ship | 219 | 84 | 2.85 | 2.3 | 61.1 | 61.7 |
| Trail riding | 679 | 90 | 3.15 | 0.8 | 59.7 | 60.7 |
| Daylily | 391 | 87 | 3.25 | 1.3 | 59.4 | 59.5 |
| Grapefruit | 506 | 83 | 3.06 | 1.0 | 59.4 | 60.4 |
| Bilberry | 228 | 75 | 3.77 | 2.2 | 58.9 | 55.2 |
| Smoothie | 330 | 78 | 3.22 | 1.5 | 58.0 | 59.8 |
| Embroidery | 356 | 81 | 3.41 | 1.5 | 57.6 | 57.2 |
| Deacon | 341 | 80 | 2.80 | 1.5 | 57.1 | 57.9 |
| Shooting range | 189 | 57 | 3.06 | 2.6 | 56.3 | 55.6 |
| Glider | 393 | 82 | 3.94 | 1.3 | 55.8 | 57.6 |
| White-tailed deer | 238 | 87 | 3.24 | 2.2 | 55.8 | 55.9 |
| Coral reef fish | 434 | 90 | 3.07 | 1.3 | 54.8 | 54.9 |
| Chevrolet silverado | 115 | 62 | 4.82 | 4.3 | 54.1 | 54.6 |
| Lugger | 103 | 62 | 3.14 | 4.9 | 53.8 | 53.8 |
| Pancit | 385 | 86 | 3.16 | 1.3 | 52.8 | 53.1 |
| Chess | 740 | 83 | 3.39 | 0.7 | 51.9 | 50.9 |
| Bakmi | 191 | 76 | 3.34 | 2.6 | 51.8 | 51.2 |
| Kitchen knife | 360 | 63 | 3.52 | 1.5 | 50.9 | 53.9 |
| Straw | 547 | 65 | 2.85 | 1.0 | 50.3 | 51.0 |
| Ancient roman architecture | 589 | 76 | 3.34 | 0.8 | 48.5 | 47.1 |
| Lamian | 257 | 80 | 3.57 | 1.9 | 47.8 | 48.2 |
| Antenna | 545 | 73 | 3.66 | 1.0 | 47.4 | 48.0 |
| Calabaza | 870 | 82 | 3.15 | 0.6 | 46.0 | 45.8 |
| Ring | 676 | 75 | 3.87 | 0.7 | 45.2 | 45.4 |
| Ciconiiformes | 426 | 88 | 3.47 | 1.2 | 45.2 | 45.2 |
| Log cabin | 448 | 70 | 3.62 | 1.1 | 44.9 | 45.7 |
| Bowed string instrument | 728 | 78 | 3.05 | 0.7 | 44.4 | 44.7 |
| Pasta | 954 | 91 | 3.21 | 0.5 | 43.7 | 43.8 |
| Knitting | 409 | 71 | 3.10 | 1.3 | 43.5 | 42.8 |
| Rope | 618 | 59 | 3.48 | 0.8 | 43.0 | 42.8 |

Table 7: **Middle $\frac{1}{3}$ of classes from Openimages for active search.** (2 of 3) Recall (%) of positives and measurements of the largest component (LC) for each selected class (153 total) from OpenImages with a labeling budget of 2,000 examples. Classes are ordered based on MLP-SEALS.

| Display Name | Total Positives | Size of the LC (%) | Average Shortest Path in the LC | Random (All) | MLP (SEALS) | MLP (All) |
|---|---|---|---|---|---|---|
| Formula racing | 351 | 88 | 3.38 | 1.4 | 42.6 | 41.4 |
| Paddy field | 468 | 70 | 4.02 | 1.1 | 42.6 | 44.2 |
| Engine | 656 | 82 | 3.43 | 0.8 | 41.7 | 40.6 |
| Electric piano | 345 | 56 | 4.15 | 1.5 | 40.9 | 42.1 |
| Shrimp | 907 | 85 | 3.82 | 0.6 | 40.4 | 40.8 |
| Goat | 1190 | 88 | 3.72 | 0.4 | 39.6 | 39.6 |
| Chocolate truffle | 288 | 58 | 5.47 | 1.8 | 39.6 | 39.9 |
| Cupboard | 898 | 88 | 3.41 | 0.6 | 39.6 | 39.6 |
| Citrus | 796 | 65 | 3.34 | 0.7 | 39.3 | 39.6 |
| Parrot | 1546 | 89 | 2.85 | 0.4 | 39.2 | 38.8 |
| Delicatessen | 196 | 52 | 2.80 | 2.6 | 38.2 | 39.0 |
| Berry | 874 | 82 | 3.78 | 0.6 | 37.8 | 37.6 |
| Briefs | 539 | 78 | 3.68 | 1.0 | 37.1 | 37.2 |
| Concert dance | 357 | 61 | 3.91 | 1.4 | 36.6 | 36.1 |
| Modern pentathlon | 772 | 43 | 2.59 | 0.6 | 35.9 | 32.6 |
| Fortification | 287 | 66 | 3.96 | 1.7 | 35.7 | 37.6 |
| Stallion | 598 | 70 | 3.58 | 0.9 | 35.7 | 36.3 |
| Belt | 467 | 41 | 3.26 | 1.1 | 35.2 | 34.9 |
| Sirloin steak | 297 | 60 | 4.97 | 1.8 | 33.9 | 32.7 |
| Stele | 450 | 70 | 3.74 | 1.1 | 33.9 | 32.7 |
| Galliformes | 674 | 82 | 3.98 | 0.7 | 33.9 | 33.9 |
| Algae | 426 | 57 | 4.52 | 1.2 | 33.8 | 33.1 |
| Herd | 648 | 75 | 3.88 | 0.8 | 33.5 | 33.7 |
| Pelecaniformes | 457 | 85 | 3.96 | 1.1 | 33.4 | 37.5 |
| Cactus | 377 | 51 | 4.11 | 1.3 | 33.4 | 35.2 |
| Shelving | 810 | 66 | 3.41 | 0.7 | 33.2 | 33.3 |
| Drums | 741 | 69 | 3.30 | 0.7 | 32.9 | 32.7 |
| Cranberry | 450 | 63 | 4.10 | 1.2 | 32.9 | 33.7 |
| Factory | 333 | 61 | 5.59 | 1.5 | 32.0 | 31.7 |
| Costume design | 818 | 52 | 3.44 | 0.6 | 30.9 | 30.6 |
| Optical instrument | 649 | 79 | 3.91 | 0.8 | 30.3 | 32.8 |
| Construction | 515 | 63 | 4.99 | 1.0 | 30.1 | 31.1 |
| Temperate coniferous forest | 328 | 59 | 4.23 | 1.5 | 30.1 | 27.6 |
| Skating | 561 | 77 | 4.04 | 1.0 | 28.8 | 30.4 |
| Egg (Food) | 1193 | 85 | 4.31 | 0.4 | 28.8 | 28.6 |
| Steamed rice | 580 | 75 | 4.54 | 0.9 | 28.1 | 30.2 |
| Plumbing fixture | 2124 | 89 | 3.19 | 0.3 | 27.9 | 27.9 |
| Whole food | 708 | 73 | 3.66 | 0.7 | 27.7 | 27.5 |
| Boardsport | 673 | 62 | 4.08 | 0.8 | 26.8 | 26.5 |
| Pork | 464 | 64 | 4.44 | 1.1 | 26.3 | 26.6 |
| Aerial photography | 931 | 63 | 3.99 | 0.6 | 25.8 | 26.1 |
| Town square | 617 | 58 | 3.69 | 0.8 | 25.7 | 26.1 |
| Estate | 667 | 51 | 4.03 | 0.9 | 24.8 | 25.9 |
| Maple | 2301 | 90 | 4.19 | 0.2 | 24.3 | 24.4 |
| Cattle | 5995 | 93 | 3.22 | 0.1 | 23.8 | 23.6 |
| Superhero | 968 | 58 | 5.28 | 0.6 | 23.4 | 23.3 |
| Bracelet | 770 | 46 | 4.13 | 0.6 | 23.2 | 24.8 |
| Frost | 483 | 60 | 4.73 | 1.0 | 23.1 | 22.5 |
| Scale model | 667 | 45 | 5.64 | 0.8 | 22.9 | 23.7 |
| Plateau | 452 | 37 | 3.88 | 1.1 | 22.7 | 19.1 |
| Bird of prey | 712 | 78 | 3.81 | 0.7 | 22.4 | 22.0 |

Table 8: **Bottom $\frac{1}{3}$ of classes from Openimages for active search.** (3 of 3) Recall (%) of positives and measurements of the largest component (LC) for each selected class (153 total) from OpenImages with a labeling budget of 2,000 examples. Classes are ordered based on MLP-SEALS.

| Display Name | Total Positives | Size of the LC (%) | Average Shortest Path in the LC | Random (All) | MLP (SEALS) | MLP (All) |
|---|---|---|---|---|---|---|
| Canal | 726 | 62 | 4.78 | 0.7 | 22.4 | 20.9 |
| Exhibition | 513 | 40 | 3.87 | 1.0 | 21.9 | 23.1 |
| Carpet | 644 | 50 | 6.98 | 0.8 | 21.9 | 22.7 |
| Monoplane | 756 | 81 | 4.70 | 0.7 | 21.8 | 20.1 |
| Ice | 682 | 50 | 4.87 | 0.8 | 21.6 | 23.1 |
| Fur | 834 | 42 | 4.31 | 0.6 | 21.2 | 17.3 |
| Icing | 1118 | 74 | 4.20 | 0.4 | 20.5 | 20.1 |
| Flooring | 814 | 38 | 3.87 | 0.6 | 20.4 | 16.9 |
| Icon | 186 | 15 | 3.26 | 2.7 | 19.9 | 17.2 |
| Prairie | 792 | 44 | 3.92 | 0.6 | 19.0 | 19.2 |
| Tooth | 976 | 49 | 4.77 | 0.5 | 18.6 | 18.0 |
| Skateboarding Equipment | 862 | 57 | 5.92 | 0.6 | 18.1 | 19.3 |
| Automotive exterior | 1060 | 23 | 2.74 | 0.5 | 17.7 | 11.9 |
| Cottage | 670 | 51 | 4.13 | 0.7 | 17.6 | 17.3 |
| Soldier | 1032 | 74 | 3.80 | 0.5 | 17.3 | 16.8 |
| Marine mammal | 2954 | 91 | 3.58 | 0.2 | 17.3 | 17.2 |
| Tool | 1549 | 64 | 4.51 | 0.3 | 17.0 | 16.9 |
| Multimedia | 741 | 46 | 4.12 | 0.7 | 16.8 | 17.1 |
| American shorthair | 2084 | 94 | 3.32 | 0.3 | 16.5 | 16.7 |
| Asphalt | 1026 | 40 | 4.53 | 0.5 | 15.1 | 11.5 |
| Singer | 604 | 56 | 4.06 | 0.9 | 14.6 | 13.6 |
| Floodplain | 567 | 50 | 4.81 | 0.9 | 14.6 | 14.0 |
| Rural area | 921 | 41 | 4.63 | 0.6 | 14.2 | 13.2 |
| Mitsubishi | 511 | 37 | 5.14 | 1.0 | 12.6 | 11.8 |
| Organ (Biology) | 1156 | 25 | 3.80 | 0.5 | 12.1 | 15.9 |
| Paper | 969 | 23 | 3.18 | 0.5 | 12.0 | 14.8 |
| Annual plant | 677 | 38 | 6.07 | 0.7 | 11.8 | 10.7 |
| Electric blue | 1180 | 19 | 3.70 | 0.5 | 11.5 | 9.4 |
| Stadium | 1654 | 77 | 5.77 | 0.3 | 10.8 | 9.3 |
| Mural | 649 | 41 | 5.24 | 0.8 | 10.4 | 10.3 |
| Teal | 975 | 16 | 4.08 | 0.5 | 9.9 | 10.4 |
| Cirque | 347 | 29 | 5.77 | 1.5 | 9.9 | 9.8 |
| Wall | 1218 | 27 | 3.13 | 0.4 | 9.3 | 12.0 |
| Thumb | 895 | 26 | 4.18 | 0.6 | 9.3 | 13.8 |
| Landscaping | 789 | 32 | 4.71 | 0.7 | 9.2 | 9.3 |
| Vehicle registration plate | 5697 | 76 | 5.89 | 0.1 | 8.7 | 8.3 |
| Meal | 1250 | 60 | 5.68 | 0.4 | 8.5 | 9.1 |
| Wilderness | 1225 | 30 | 4.12 | 0.4 | 8.5 | 9.8 |
| Liqueur | 539 | 51 | 5.98 | 1.0 | 8.0 | 12.8 |
| Space | 1006 | 23 | 4.63 | 0.5 | 7.8 | 6.3 |
| Cycling | 794 | 63 | 5.00 | 0.6 | 7.3 | 7.8 |
| Brown | 1427 | 16 | 3.49 | 0.4 | 7.2 | 2.6 |
| Organism | 1148 | 21 | 3.49 | 0.4 | 6.8 | 2.0 |
| Laugh | 750 | 19 | 6.22 | 0.7 | 6.6 | 8.6 |
| Bumper | 985 | 37 | 6.65 | 0.5 | 5.9 | 8.3 |
| Portrait | 2510 | 67 | 6.38 | 0.2 | 5.8 | 5.3 |
| Mode of transport | 1387 | 24 | 4.50 | 0.4 | 5.1 | 3.6 |
| Interaction | 924 | 15 | 6.05 | 0.6 | 4.5 | 4.6 |
| Tournament | 841 | 47 | 9.90 | 0.6 | 4.3 | 5.1 |
| Performing arts | 1030 | 29 | 6.97 | 0.5 | 2.3 | 2.5 |
| White | 1494 | 3 | 2.79 | 0.3 | 2.0 | 0.5 |

## A.8    Self-supervised embedding (Sentence-BERT) on Goodreads

We followed the same general procedure described in Section 4.1, aside from the dataset specific details below. Goodreads spoiler detection (Wan et al., 2019) had 17.67 million sentences with binary spoiler annotations. Spoilers made up 3.224% of the data, making them much more common than the rare concepts we evaluated in the other datasets. Following Wan et al. (2019), we used 3.53 million sentences for testing (20%), 10,000 sentences as the validation set, and the remaining 14.13 million sentences as the unlabeled pool. We also switched to the area under the ROC curve (AUC) as our primary evaluation metric for active learning to be consistent with Wan et al. (2019). For $G_z$, we used a pre-trained Sentence-BERT model (SBERT-NLI-base) (Reimers & Gurevych, 2019), applied PCA whitening to reduce the dimension to 256, and performed $l^2$ normalization.

### A.8.1    Active search

SEALS achieved the same recall as the baseline approaches, but only considered less than 1% of the unlabeled data in the candidate pool, as shown in Figure 15. At a labeling budget of 2,000, MLP-ALL and MLP-SEALS recalled $0.15 \pm 0.02\%$ and $0.17 \pm 0.05\%$, respectively, while MaxEnt-All and MaxEnt-SEALS achieved $0.14 \pm 0.04\%$ and $0.11 \pm 0.06\%$ recall respectively. Increasing the labeling budget to 50,000 examples, increased recall to ~3.7% for MaxEnt and MLP but maintained a similar relative improvement over random sampling, as shown in Figure 16. ID-SEALS performed worse than the other strategies. However, all of the active selection strategies outperformed random sampling by up to an order of magnitude.

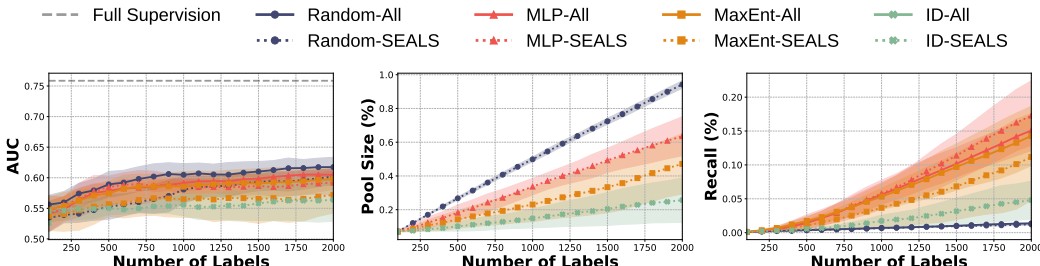

Figure 15: Active learning and search on Goodreads with Sentence-BERT embeddings. Across datasets and strategies, SEALS with $k = 100$ performs similarly to the baseline approach in terms of both the error the model achieves for active learning (left) and the recall of positive examples for active search (right), while only considering a fraction of the data $U$ (middle).

### A.8.2    Active learning

At a labeling budget of 2,000 examples, all the selection strategies were indistinguishable from random sampling. Increasing the labeling budget did not help, as shown in Figure 16. Unlike ImageNet and OpenImages, Goodreads had a much higher fraction of positive examples (3.224%), and the examples were not tightly clustered as described in Section A.8.3.

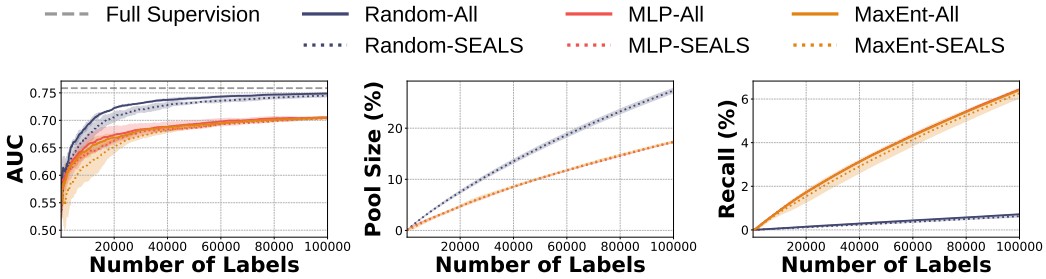

Figure 16: Active learning and search on Goodreads with a labeling budget of 100,000 examples. Across strategies, SEALS with $k = 100$ performed similarly to the baseline approach in terms of both the error the model achieved for active learning (left) and the recall of positive examples for active search (right), while only considering a fraction of the data $U$ (middle). ID was excluded because of the growing pool size and computation. For active search, MaxEnt and MLP continued to improve recall. For active learning, all the selection strategies (both with and without SEALS) performed worse than random sampling despite the larger labeling budget. This gap was likely due to spoilers being book specific and the higher fraction of positive examples in the unlabeled pool, causing relevant examples to be spread almost uniformly across the space (see Section A.8.3).

### A.8.3  LATENT STRUCTURE

The large number of positive examples in the Goodreads dataset limited the analysis we could perform. We could only calculate the size of the largest connected component in the nearest neighbor graph (Figure 17). For $k = 10$, only 28.4% of the positive examples could be reached directly, but increasing $k$ to 100 improved that dramatically to 96.7%. For such a large connected component, one might have expected active learning to perform better in Section A.8.2. By analyzing the embeddings, however, we found that examples are spread almost uniformly across the space with an average cosine similarity of 0.004. For comparison, the average cosine similarity for concepts in ImageNet and OpenImages was $0.453 \pm 0.077$ and $0.361 \pm 0.105$ respectively. This uniformity was likely due to the higher fraction of positive examples and spoilers being book specific while Sentence-BERT is trained on generic data. As a result, even if spoilers were tightly clustered within each book, the books were spread across a range of topics and consequently across the embedding space, illustrating a limitation and opportunity for future work.

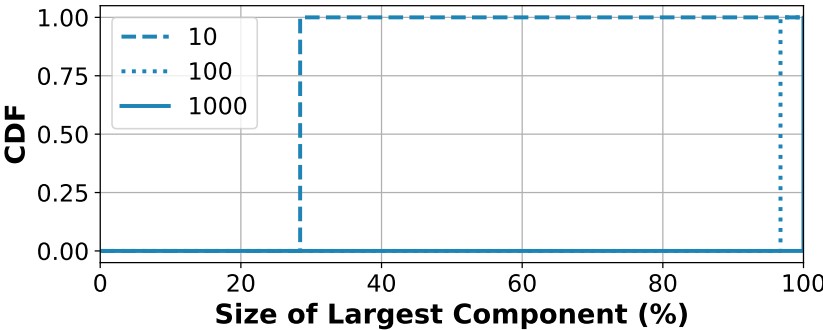

Figure 17: Cumulative distribution function (CDF) for the largest connected component in the Goodreads dataset with varying values of $k$.

