# OpenReview forum: "Similarity Search for Efficient Active Learning and Search of Rare Concepts"
_ICLR.cc/2021/Conference — Reject_

### Official Review · AnonReviewer3 · 2020-10-27
**Accept - strong motivation and results**

**Rating:** 8
**Confidence:** 3

**Review:**

This paper proposes an active learning and active search approach that targets samples for rare classes in very large unlabeled datasets with highly imbalanced class distributions. This is a common scenario in real-world applications, where these rare situations can be critical to accurately categorize - ie endangered species. The authors propose an approach that targets these rare cases while reducing the number of overall examples sampled, and that scales with the amount of labeled data as opposed to the amount of unlabeled data which allows them to consider datasets up to billions of examples.

Pros:

This is a well-motivated task, there is a clear need for this type of method as access to unlabeled data increases
Their method scales effectively to very large sets of unlabeled data, eg matching baseline performance with only 0.1% of the unlabeled data sampled on their proprietary 10 billion image dataset.
The method leads to impressive computational speedups in active learning selection rounds, eg 4000x speedup in selection round time on ImageNet.

Cons -

I am glad to see that they compare performance to public datasets to help with this, but some of their biggest claims are not reproducible because they are shown with proprietary data.
Their model does require an initial pass through an embedding model for all the data, which would be very slow and expensive for 10 billion images. This embedding model seems to need to encode the rare classes appropriately and be able to cluster them together sufficiently for the neighbors of a rare class to be likely to be of the same class for this method to work well. In the low data regime it is not always guaranteed that you will have sufficient training data to build an embedding that handles rare categories sensibly, particularly in fine-grained scenarios. The authors could do a better job analyzing the impact of their chosen embedding function on the efficacy of their method. They discuss this a bit in section 4.4, but it would be awesome to see some concept of performance of the method for each concept included in Figure 3 (unsure how best to do this).

---

> ### Author Response · Authors · 2020-11-18
> **Added a more detailed discussion**
>
> Thank you for your thoughtful feedback! We addressed many of your points in the updated draft and will continue to revise the paper during the rebuttal period. To begin the discussion, we added a more detailed discussion about creating the embeddings and index in Section 3.3. For regimes with few labels and no auxiliary tasks to transfer from, self-supervision over the unlabeled data can be used to build the embeddings. Empirically, we observed similar results with self-supervised embeddings using SimCLR (Chen et al., 2020) for ImageNet (Appendix A.3) as the results presented in the main text. These experiments also demonstrate the impact the embedding function can have.
>
> We worked on figures to show our method's performance for each concept included in Figure 3 but found it difficult to detangle the impact of SEALS from the ease of the classification problem. The more tightly clustered the concept is, the easier it is to classify in general, so plotting performance vs. largest connected component was not incredibly insightful. We will continue to iterate.

---

> > ### Author Response · Authors · 2020-11-24
> > **Added Figure 11 to show the impact of the latent structure on SEALS**
> >
> > We added Figure 11 to show the impact of the embedding and the resulting k-NN graph. In general, SEALS performed better for concepts that formed larger connected components and had shorter paths between examples.

---

### Official Review · AnonReviewer4 · 2020-10-27
**Novelty is limited, and lack of theoretical analysis**

**Rating:** 4
**Confidence:** 4

**Review:**

Summary:
This paper proposes a new method (SEALS) to accelerate the active learning and active search with the skewness of the cardinality of rare class compared to the large-scale datasets. To leverage this skewness, the authors restrict the candidate pool for labelling mainly from the nearest neighbours of the currently labelled set (except the initial set). The authors conduct very detailed experiments on the tasks of active learning and active search over three large-scale data sets to validate the efficiency and effectiveness of SEALS.

Pros:
1. The motivation to leverage the skewness of data to speed up active learning and active search is good. I like the idea to incrementally enlarge the candidate pool for labelling by the nearest neighbours.
2. The experiment is rigorous and the authors also consider and provide results on NLP domain in the supplementary.
3. The method SEALS seems to be general and can be adapted to different active learning methods of rare concepts.

Cons:
1. The major concern of this paper is the lack of theoretical analysis. The idea is straightforward and the novelty is limited. The authors use very detailed experiments to verify their observation, but is it technically solid? Does this method require any (strong) assumption? Can you get any novel theoretical findings?  A sentence about ``learned representations can effectively cluster rare concepts” is not enough.
2. There exist many errors and over claims in the statement of k-Nearest Neighbour Search (k-NNS). More details can be found in minor comments.
3. The time complexity of information density (ID) and its implementation in the experiments for comparison are wrong. With some pre-computation, the time complexity of ID should be as same as the max entropy (MaxEnt). More details can also be found in minor comments.
4. The Introduction section only discusses the motivation of active learning for rare concepts. How about the active search? Is there any more technique challenge, or just conduct one more task only? I have this concern because the objective function of MLP is quite different from MaxEnt and ID, which is not necessary to validate the idea of using the skewness for acceleration.

Minor Comments or Typos:
1. The authors claim that k-NNS can be done in logarithmic time [1] or approximately in constant time [2][3]. As a researcher whose research areas include this, I would like to point out that this claim is wrong. For the kd-tree [1], the log time only happens when the data dimension is restricted to 2 or 3. For the scenarios the authors consider in this paper, the data dimension is at least 100 dimensions, the data structure of kd-tree will suffer from the curse of dimensionality. Moreover, the authors of SimHash [2] and FAISS [3] did not claim their methods can handle k-NNS approximately in constant time. In fact, there is no method with a theoretical guarantee that can deal with (approximate) k-NNS in O(log n) time in high-dimensional Euclidean spaces (e.g., d > 100). Specifically, the time complexity of SimHash [2] is sub-linear, i.e., O(\log(n) \cdot n^\rho) with \rho \in (0,1), which is still far from log(n) level.
2. The authors also claim they use locality-sensitive hashing (LSH) [2] with Euclidean distance implemented in FAISS [3]. I would like to point out SimHash [3] is designed for cosine similarity instead of Euclidean distance. And FAISS is built on Product Quantization instead of LSH. I guess the authors only use FAISS in the experiments to determine nearest neighbours with Euclidean distance, but this sentence is misleading.
3.  As discussed in [4], the density score (e.g., the sum of the similarity for each data) can be pre-computed and cached for efficient lookup. So the time complexity of ID in Table 1 should be as same as MaxEnt. And similarly, the time complexity of SEALS can be reduced to O(k^2 |L_r|). Thus, in the experiments, the author can pre-compute the density score and hence the ID strategy can be evaluated in the private ten billion data set.
4. The notation of \mathcal{L}_r in Algorithms 1 & 2 is inconsistent with the paper which is L_r.
5. In line 8 of Algorithm 2, remove “-z*” as follows
\mathcal{P}_r = \mathcal{P}_r \cup \mathcal(N)(z*,k) – z* --> \mathcal{P}_r = \mathcal{P}_r \cup \mathcal(N)(z*,k)
6. In the experiments, it will be better to also report the running time besides the pool size for active search.

Reference:

[1] Bentley, Jon Louis. "Multidimensional binary search trees used for associative searching." Communications of the ACM 18, no. 9 (1975): 509-517.

[2] Charikar, Moses S. "Similarity estimation techniques from rounding algorithms." In Proceedings of the thirty-fourth annual ACM symposium on Theory of computing, pp. 380-388. 2002.

[3] Johnson, Jeff, Matthijs Douze, and Hervé Jégou. "Billion-scale similarity search with GPUs." IEEE Transactions on Big Data (2019).

[4] Settles, Burr, and Mark Craven. "An analysis of active learning strategies for sequence labeling tasks." In Proceedings of the 2008 Conference on Empirical Methods in Natural Language Processing, pp. 1070-1079. 2008.

==============================================================================================

Update: Thank you for the feedback and the efforts in revising the draft. Some of my questions were clarified and many existing errors have been fixed. However, I still think the novelty is limited, and more needs should be done to enrich the method with more analysis in terms of theoretical and mathematical aspects. Based on the above reasons, I do not intend to increase my rating.

---

> ### Author Response · Authors · 2020-11-18
> **Added more technical details and the key takeaways still hold.**
>
> Thank you for your thoughtful feedback! We addressed many of your points in the updated draft and will continue to revise the paper during the rebuttal period. To begin the discussion, we made the following high-level changes:
>
> - Corrected the claims about k nearest neighbors (k-NN)
> - Replaced Table 1 with a more nuanced explanation of computational complexity in Section 3.3 and a detailed description of our implementation choices in Section 4.1.
> - Clarified our explanation of information density to point out that the average similarity scores for each example were cached as in original work (Settles & Craven, 2008).
> - Added a new Table with the wall clock runtimes for varying selection strategies on ImageNet and OpenImages (now Table 2). The total runtime is also broken down into 3 parts: 1) the time to apply the selection strategy to the candidate pool, 2) the time to find the k nearest neighbors (k-NN) for the newly labeled examples, and 3) the time to train logistic regression on the currently labeled examples.
>
> Concerning novelty, we view the simplicity of the SEALS approach as a strength rather than a weakness. It can be applied transparently for many selection strategies making it applicable to a wide range of active learning and search methods, even beyond the ones considered here. The computational savings are substantial and complementary to other techniques for improving efficiency, such as batch active learning and using proxy models during selection, making existing active learning strategies much more scalable. This approach also introduces another use case of similarity search, representing an exciting avenue for future work.  In particular, we motivate the problem of doing active learning and active search for rare concepts at massive scale, which is both practically and theoretically interesting but has not been heavily studied in research. Most public datasets used for active learning and search do not have the scale or structure to reflect problems we experience in industry, where the concept of interest is extremely scarce (less than 1 in 1,000 or more). We propose SEALS as a strong baseline for future work in this area.
>
> While a rigorous theoretical argument is outside of the rebuttal's scope, we can provide an intuitive explanation for how SEALS works. By viewing the datasets through their k-NN graphs, we empirically found that many concepts formed large connected components (Section 4.4). This observation simplifies active search to a graph traversal problem, where we want to find the boundaries of the connected components. MLP-SEALS can be seen as a variant of the flood-fill algorithm. Rather than a simple depth-first search, MLP greedily navigates the graph to minimize the number of examples that need to be visited/labeled. For active learning, there are also graph-based algorithms such as the shortest shortest path algorithm (S^2) from Dasarathy et al. (Conference on Learning Theory 2015). S^2 selects examples to label by bisecting the minimum shortest shortest path (MSSP) between oppositely labeled examples and provides strong guarantees for datasets with large connected components. Intuitively, bisecting the MSSP can be viewed as a measure of informativeness or uncertainty. From this perspective, our SEALS approach is effectively replacing the MSSP procedure, which would be intractable for large datasets, with a much more computationally efficient heuristic like max entropy and lazily computing the k-NN graph.

---

> > ### Author Response · Authors · 2020-11-18
> > **Additional points**
> >
> > **pre-compute the density score and hence the ID strategy can be evaluated in the private ten billion data set.**
> >
> > There are indeed ways to speed up quadratic methods, but they are still quadratic. Caching the density scores helps subsequent rounds but pre-computing the initial scores is prohibitively expensive for large-scale datasets with millions or billions of examples. On OpenImages (6,816,296 examples), the baseline for information density ran for over 24 hours without completing the first round.
> >
> > **The notation of \mathcal{L}_r in Algorithms 1 & 2 is inconsistent with the paper which is L_r.**
> >
> > $\mathcal{L}_r$ represents the data in the embedding space, while $L_r$ represents the raw data.
> >
> > **In line 8 of Algorithm 2, remove “-z” as follows \mathcal{P}_r = \mathcal{P}_r \cup \mathcal(N)(z,k) – z* --> \mathcal{P}_r = \mathcal{P}_r \cup \mathcal(N)(z*,k)**
> >
> > $\mathbf{-z}$ was meant to show that the newly labeled example was removed from the candidate pool. We changed this to $\mathcal{P}_r =( \mathcal{P}_r \setminus \{ \mathbf{z^*} \}) \cup \mathcal{N}(\mathbf{z^*},k)$ to make this clearer.

---

### Official Review · AnonReviewer1 · 2020-11-01
**Adopt k-NN to enhance active learning efficiency w/ solid experiments but w/o technical depths**

**Rating:** 5
**Confidence:** 5

**Review:**

Summary
-------

This paper adopts k-NN to enhance the efficiency of active learning for heavily skewed data sets up to 10 billion scale. The proposed algorithm is evaluated extensively by a large number of experiments. The experimental results show the efficiency of proposed algorithm with 10x times speedup comparing to existing methods.

The proposed algorithm is presented in the form of pseudo codes (Algo 2), but without clear introduction to the details of the Algo 2, particularly, the detail of how the k-NN will be performed, and the analysis of the cost model. This makes the paper lack of technical depth.


Strengths
---------

- The motivation of this work is clear. The authors emphasize the motivation and difference from existing again and again in the whole paper. The paper is well-written and well-structed thus easy for understanding.

- A 10 billion-scale data set with heavy skew is used for experiments, which make the evaluations convincing (meanwhile it could be counted as a weakness as well. see next section).

- The extensive experiments are sufficient to show the advantages of the proposed algorithm for active learning and active search.

Weaknesses
----------

- The approach of adopting k-NN to enhance the efficiency of data sampling during the active learning and search is a natural combination with active learning framework, thus there is no surprising at the view point of novelty. The reasons of such judgment are based on the following points.

	- The authors only introduce the advantages of k-NN that can be done in logarithmic time or even in constant time for approximate k-NN if adopting latest third-party implementations. However, the authors did not introduce how non-trivial the idea of adopting k-NN is, or how difficult the authors came up with such a idea. It means that currently the proposed method sound only a natural combined algorithm but without high novelty.

	- The key contribution should be how to incorporate k-NN with active learning and active search in this paper. Actually, such kind of incorporation has been studied and proposed in some key publications. Thus the authors have to present the originality of their approach.

    [1] Kai Wei, et al.: Submodularity in Data Subset Selection and Active Learning (ICML 2015)
    [2] Ajay J. Joshi, et al.: Coverage Optimized Active Learning for K-NN Classifiers (ICRA 2012)

- The technical depth of proposed algorithm is not sufficient. For example, the authors could add more details to introduce how the k-NN algorithm is collaborating with active learning in terms of theoretical and mathematical aspects. However, the authors only put the algorithm in the paper with a table to show the comparative computation complexity. More details and clarifications should be provided to make the approach in a technical way.

- The 10 billion data set is very big and should benefit the research community if it will be open-sourced or available for public use. However, it sounds a data set from a large internet company and involving with hired annotators during the experiments. So, there might be some unseen factors that would have more or less impact on the evaluation results. This point should be clarified by the authors.


Other Questionable Points
-------------------------

- Sec 3.1: each unlabeled data $x_i$ is mapped to a latent variable $G_z(x_i) = z_i$. Here, as we know, high dimensional features (i.e., $z_i$) is often used in many computation vision tasks. In such cases, the k-NN computation itself will get involved into the well-known problem -- "the curse of dimensionality", that makes k-NN algorithm cannot be done in logarithmic time but turn to linear time. Therefore, the basic assumption that k-NN computation can be done efficiency does not hold any more. This critical point is not discussed in the paper.

- Table 1: the complexity of SEALS might be incorrect. As a reviewer, if my understanding is correct, the computation of k-NN ($N(z,k)$) should involve the similarity computations between labeled data ($z$) and unlabeled data ($o \in U$). Although the authors assume employing efficient indexing method to perform k-NN computation, its complexity should be counted as $O(log|U|)$ as well. It will become $O(|U|)$ in the worst cases when the curse of dimensionality happened.

- Table 2: Why the fraction positive regarding the 10B image dataset is not available in the table?

- Fig. 1(b): Why the performance of mAP regarding MLP-x, MaxEnt-x methods has drops at the beginning when the number of labels is less then 500?

---

> ### Author Response · Authors · 2020-11-18
> **Added additional comparisons to related work and more technical details**
>
> Thank you for your thoughtful feedback! We addressed many of your points in the updated draft and will continue to revise the paper during the rebuttal period. To begin the discussion, we made the following high-level changes:
>
> - Added an additional paragraph to the related work section to clarify our work's novelty compared to the prior work you mentioned. Additional details below.
> - Replaced Table 1 with a more nuanced explanation of computational complexity in Section 3.3 and a detailed description of our implementation choices in Section 4.1.
> - Added a new Table with the wall clock runtimes for varying selection strategies on ImageNet and OpenImages (now Table 2). The total runtime is also broken down into 3 parts: 1) the time to apply the selection strategy to the candidate pool, 2) the time to find the k nearest neighbors (k-NN) for the newly labeled examples, and 3) the time to train logistic regression on the currently labeled examples.
>
> Concerning technical depth, we argue the simplicity of the SEALS approach is a strength rather than a weakness. It can be applied transparently for many selection strategies making it applicable to a wide range of active learning and search methods, even beyond the ones considered here. The computational savings are substantial and complementary to other techniques for improving efficiency, such as batch active learning and using proxy models during selection, making existing active learning strategies much more scalable. This approach also introduces another use case of similarity search, representing an exciting avenue for future work. In particular, we motivate the problem of doing active learning and active search for rare concepts at massive scale, which is both practically and theoretically interesting but has not been heavily studied in research. Most public datasets used for active learning and search do not have the scale or structure to reflect problems we experience in industry, where the concept of interest is extremely scarce (less than 1 in 1,000 or more). We propose SEALS as a strong baseline for future work in this area.
>
> While a rigorous theoretical argument is outside of the rebuttal's scope, we can provide an intuitive explanation for how SEALS works. By viewing the datasets through their k-NN graphs, we empirically found that many concepts formed large connected components (Section 4.4). This observation simplifies active search to a graph traversal problem, where we want to find the boundaries of the connected components. MLP-SEALS can be seen as a variant of the flood-fill algorithm. Rather than a simple depth-first search, MLP greedily navigates the graph to minimize the number of examples that need to be visited/labeled. For active learning, there are also graph-based algorithms such as the shortest shortest path algorithm (S^2) from Dasarathy et al. (Conference on Learning Theory 2015). S^2 selects examples to label by bisecting the minimum shortest shortest path (MSSP) between oppositely labeled examples and provides strong guarantees for datasets with large connected components. Intuitively, bisecting the MSSP can be viewed as a measure of informativeness or uncertainty. From this perspective, our SEALS approach is effectively replacing the MSSP procedure, which would be intractable for large datasets, with a much more computationally efficient heuristic like max entropy and lazily computing the k-NN graph.
>
> We cannot open source the 10 billion images dataset due to privacy constraints, but the OpenImages dataset is publicly available and serves as a reasonable proxy. It is already about 2 orders of magnitude larger than other datasets used in the active learning literature, which typically only have tens of thousands of examples. The largest dataset in Wei et al. (ICML 2015) and Joshi et al. (ICRA 2012) was MNIST with 60,000 examples, while OpenImages had 6,816,296 examples at the time of writing.

---

> > ### Author Response · Authors · 2020-11-18
> > **Additional details**
> >
> > **Comparison to other works that incorporate k-NN with active learning**
> >
> > Wei et al. (ICML 2015) and Joshi et al. (ICRA 2012) trained k-NN classifiers and used selection strategies that scored the entire unlabeled dataset in each round. Both employed a similar two-step approach. First, they select a large batch of the most uncertain examples to incorporate informativeness. Then, they select a smaller subset based on a submodular utility function to ensure representativeness. The first step still requires a linear pass over the unlabeled data described in line 4 of Algorithm 1 from Wei et al.’s work and the end of Section V in Joshi et al.’s work. Joshi et al. additionally uses LSH but purely as a way to speed-up the k-NN classifier. In comparison, we use k-NN search to create and expand the candidate pool. We do not formulate or use a k-NN classifier at any point. This is an important, but subtle difference that is novel and complementary to prior work that combines k-NN classifiers with active learning.
> >
> > **Why is the fraction positive regarding the 10B image dataset is not available in the table?**
> >
> > The 10B images dataset is unlabeled, so we do not know the ground truth for how frequently the concepts appear in the data. Examples are labeled as they are selected by the active learning and search strategies in real-time, making the observed labels heavily biased.

---

### Decision · Program_Chairs · 2021-01-07
**Final Decision**

**Decision:**

Reject

**Comment:**

The paper proposed an active search algorithm for efficiently identifying rare concepts among heavily imbalanced datasets. Reviewers find the paper very well-motivated and addressing an important real-world challenge in active learning. All reviewers appreciate the extensive demonstration of the effectiveness of the proposed algorithm on real-world tasks, in particular in industrial settings where the scale of problems goes far beyond common academic datasets.

In the meantime, there are shared concerns among several reviewers in the technical depth of the proposed algorithm. Although the authors provided intuitive explanations of the nearest-neighbor based approach, the results reported are restricted mostly to several final performance metrics on the search performance. As a purely empirical work, the paper would benefit from more fine-grained experimental analyses and ablation studies (e.g., by breaking down to analyses at intermediate levels).